# LightZero: A Unified Benchmark for Monte Carlo Tree Search in General Sequential Decision Scenarios

**Yazhe Niu**[1,3]    **Yuan Pu**[2]    **Zhenjie Yang**[1]    **Xueyan Li**[2]    **Tong Zhou**[1]
**Jiyuan Ren**[2]    **Shuai Hu**[1]    **Hongsheng Li**[3,4] [*]    **Yu Liu**[1,2]

[1]SenseTime Group LTD
[2]Shanghai Artificial Intelligence Laboratory
[3]The Chinese University of Hong Kong
[4]Centre for Perceptual and Interactive Intelligence

## Abstract

Building agents based on tree-search planning capabilities with learned models has achieved remarkable success in classic decision-making problems, such as *Go* and *Atari*. However, it has been deemed challenging or even infeasible to extend Monte Carlo Tree Search (MCTS) based algorithms to diverse real-world applications, especially when these environments involve complex action spaces and significant simulation costs, or inherent stochasticity. In this work, we introduce *LightZero*, the first unified benchmark for deploying *MCTS/MuZero* in general sequential decision scenarios. Specifically, we summarize the most critical challenges in designing a general MCTS-style decision-making solver, then decompose the tightly-coupled algorithm and system design of tree-search RL methods into distinct sub-modules. By incorporating more appropriate exploration and optimization strategies, we can significantly enhance these sub-modules and construct powerful LightZero agents to tackle tasks across a wide range of domains, such as board games, *Atari*, *MuJoCo*, *MiniGrid* and *GoBigger*. Detailed benchmark results reveal the significant potential of such methods in building scalable and efficient decision intelligence. The code is available as part of OpenDILab at https://github.com/opendilab/LightZero.

## 1  Introduction

General decision intelligence needs to solve tasks in many distinct domains. Recent advances in reinforcement learning (RL) algorithms have addressed several challenging decision-making problems [1, 2] and even surpassed top-level human experts in performance [3]. However, these state-of-the-art RL agents often exhibits poor data efficiency and face significant challenges when handling a wide range of diverse problems. Different environments present specific learning requirements and difficulties that prompted currently various algorithms (e.g. DQN [4], PPO [5], R2D2 [6], SAC [7]) and system architectures such as IMPALA [8] and others [9, 10, 11]. Designing a general and data-efficient decision solver needs to tackle various challenges, while ensuring that the proposed algorithm can be universally deployed anywhere without domain-specific knowledge requirements.

Monte Carlo Tree Search (MCTS) is a powerful approach that utilizes a search tree with simulation and backpropogation mechanisms to train agents with a small data budget [12]. To model high-dimensional observation spaces and complex policy behaviour, AlphaGo [13] enhances MCTS with deep neural networks and designs the policy and value network that identify optimal actions and winning rates respectively, which was the first to defeat the strongest professional human player in Go. Despite the impressive results, MCTS-style algorithms rely on a series of necessary conditions, such as knowledge of game rules and simulators, discrete action space and deterministic state transition, which severely restrict the application scope of these methods. In recent years, several successors

---

[*]Corresponding Author

37th Conference on Neural Information Processing Systems (NeurIPS 2023) Track on Datasets and Benchmarks.

to AlphaGo have attempted to extend its capabilities in various directions. MuZero [14] relaxes the requirements for prior knowledge of environments by training a set of neural networks to reconstruct reward, value and policy. Sampled MuZero [15] successfully applies MCTS to various complex action space with a novel planning mechanism based on sampled actions. [16, 17, 18] improve MuZero in terms of planning stochasticity, representation learning effectiveness and simulation efficiency respectively. These emerging algorithm insights and techniques have contributed to the development of more general MCTS algorithms and toolchains.

In this paper, we present a unified algorithm benchmark named *LightZero* that first comprehensively integrates different MCTS/MuZero algorithm branches, including 9 algorithms and more than 20 decision environments with detailed evaluation. To better understand the potential of MCTS as an efficient general-purpose sequential decision solver, we revisit the development history of MCTS methods [19] and the diverse criterions of newly proposed RL environments [20, 21, 22]. As shown in Figure 2, we outline the six most challenging dimensions in developing LightZero as a general method, including multi-modal and high-dimensional observation space [23], complex action space, reliance on prior knowledge, inherent stochasticity, simulation cost, and hard exploration.

Furthermore, highly coupled algorithm and system architectures greatly increase the cost and barriers of migrating and improving MCTS-style methods. Some special mechanisms like tree search and data reanalyze [24] seriously hinder the simplification and parallel acceleration of code implementation. To overcome these difficulties, LightZero designs a modularly pipeline to enable distinct algorithm components as plug-ins. For example, the chance node planning for modelling stochasticity can also be used in continuous control or hybrid action environments. From the unified viewpoint provided by LightZero, we can systematically divide the whole training scheme of MCTS-style methods into four sub-modules: data collector, data arranger, agent learner, and agent evaluator. LightZero's decoupled architecture empowers developers to focus intensively on the customization of environments and algorithms. Meanwhile, some techniques like off-policy correction and data throughput limiter can ensure the steady convergence of the algorithm while achieving runtime speedups.

Based on these supports, LightZero also explores the advantages of combining some novel insights from model-based RL with MCTS approaches. In particular, the misalignment problem [25] of state representation learning and dynamics learning can result in the problematic optimization for MuZero, thus a simple self-consistency loss can significantly speed up convergence without special tuning. Besides, intrinsic reward mechanism [26] [27] [28] can address the exploration deficiency of tree-search methods with hand-crafted noises. Subsequently, we evaluate the ability of LightZero as a general solver for various decision problems. Experiments on different types of environments demonstrate LightZero's rich application ranges and data efficiency regimes with few hyper-parameter adjustments. At last, we provide discussions on the future optimization directions of each sub-module.

In general, we summarize the three key contributions of this paper as follows:

- We present LightZero, the first general MCTS/MuZero algorithm benchmark that systematically evaluates related algorithms and system designs.

- We outline the most critical challenges of real-world decision applications. To address these issues, we decouple the algorithm and system design of MCTS methods and design a modular training pipeline, which can easily integrate novel insights for better scalability.

- We demonstrate the capability and future potential of LightZero as a general sequential decision solver, which can be trained and deployed across diverse domains.

## 2   Background

**Reinforcement Learning** models a decision-making problem as a Markov Decision Process (MDP) $\mathcal{M} = (\mathcal{S}, \mathcal{A}, \mathcal{P}, \mathcal{R}, \gamma, \rho_0)$, where $\mathcal{S}$ and $\mathcal{A}$ denote the state space and action space, respectively. The transition function $\mathcal{P}$ maps $\mathcal{S} \times \mathcal{A}$ to $\mathcal{S}$, while the expected reward function $\mathcal{R}$ maps $\mathcal{S} \times \mathcal{A}$ to $\mathbb{R}$. The discount factor $\gamma \in [0, 1)$ determines the importance of future rewards, and $\rho_0$ represents the initial state distribution. The goal of RL is to learn a policy $\pi : \mathcal{S} \to \mathcal{A}$ that maximizes the expected discounted return over the trajectory distribution $J(\pi) = \mathbb{E}_{\pi, \rho_0, \mathcal{P}, \mathcal{R}}[\sum_{t=0}^{\infty} \gamma^t r_t]$.

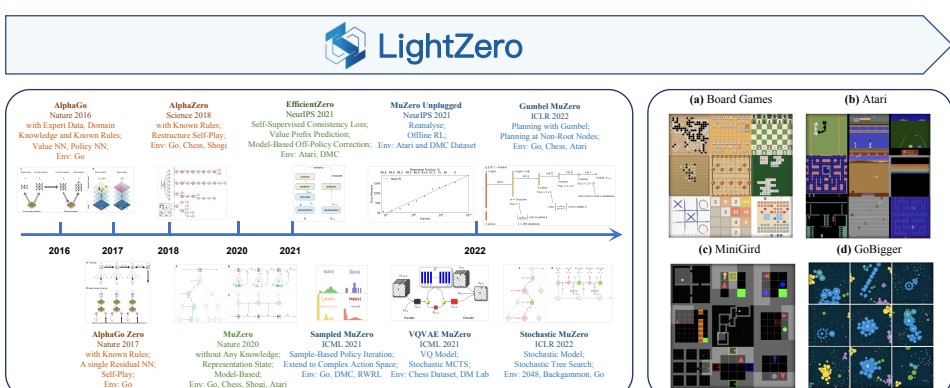

Figure 1: Overview of LightZero. The left side depicts the development of MCTS, while the right side showcases various RL environments. LightZero incorporates and extends recent advances within the MCTS/MuZero sub-domain and effectively applies them across diverse environments.

**AlphaZero** [29] is a generalized version of AlphaGo [13], eliminating the reliance on supervised learning from game records. It is trained entirely through unsupervised self-play and achieves superhuman performance in various board games, such as chess, shogi, and Go. This approach replaces the handcrafted features and heuristic priors commonly used in traditional intelligent programs. Specifically, AlphaZero employs a deep neural network parameterized by $\theta$, represented as $(\mathbf{p}, v) = f_\theta(s)$. Given a board position $s$, the network produces a action probability $p_a = P_r(a|s)$ for each action $a$ and a scalar value $v$ to predict the expected return $z$, i.e. $v \to z$.

**MuZero** [14] achieves superhuman performance in more complex domains with visual input [30], without knowledge of the environment's transition rules. It combines tree search with a learned model, using three networks: ① Representation Network: $s^0 = h_\theta(o_1, \ldots, o_t)$. This network represents the root node (at time $t$) as a latent state, obtained by processing past observations $o_1, \ldots, o_t$, ② Dynamics Network: $r^k, s^k = g_\theta(s^{k-1}, a^k)$. This network simulates the dynamics of the environment. Given a state and selected action, it outputs the transitioned next state and corresponding reward. ③ Prediction Network: $\mathbf{p}^k, v^k = f_\theta(s^k)$. Given a latent state, this network predicts the action probability and value. Notably, MuZero searches within the learned latent space. For the MCTS process in MuZero, assume the initial root node $s_0$ is generated from the original board state through the representation network, each edge stores the following information: $N(s, a), P(s, a), Q(s, a), R(s, a), S(s, a)$, respectively representing visit counts, policy, mean value, reward, and state transition. The MCTS process in the latent space can be divided into three phases:

- **Selection**: Actions are chosen according to the Upper Confidence Bound (UCB) [31] formula:

$$a^* = \arg \max_a Q(s, a) + P(s, a) \frac{\sqrt{\sum_b N(s, b)}}{1 + N(s, a)} [c_1 + \log(\frac{\sum_b N(s, b) + c_2 + 1}{c_2})]$$

  where, $N$ represents the visit count, $Q$ is the estimated average value, and $P$ is the policy's prior probability. $c_1$ and $c_2$ are constants that control the relative weight of $P$ and $Q$.

- **Expansion**: The selected action is executed in the learned model, continuing until a leaf node is encountered. At this point, a new state node $s^l$ is generated, and its associated predicted reward $r^l$ is determined. Utilizing the prediction function, we obtain the predicted values $p^l$ and $v^l$. Subsequently, this node is incorporated into the search tree.

- **Backup**: The estimated cumulative reward at step $k$ is calculated based on $v^l$, denoted as: $G^k = \sum_{\tau=0}^{l-1-k} \gamma^\tau r_{k+1+\tau} + \gamma^{l-k} v^l$. Subsequently, $Q$ and $N$ are updated along the search path.

After the search is completed, the visit count set $N(s, a)$ is returned at the root node $s_0$. These visit counts are normalized to obtain the improved policy:

$$\mathcal{I}_\pi(a|s) = N(s, a)^{1/T} / \sum_b N(s, b)^{1/T}$$

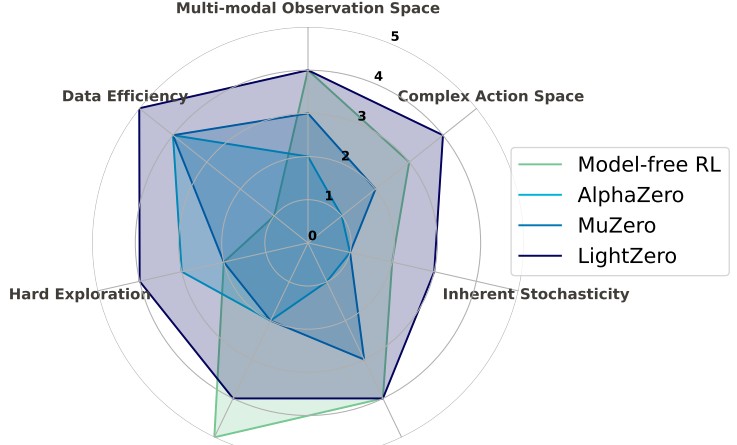

Figure 2: Radar chart comparison of MCTS-style methods and model-free RL (e.g. PPO) on six environment challenges and another data efficiency dimensions. We categorize the critical capabilities of general decision solvers as follows: multi-modal observation space, complex action space, inherent stochasticity, reliance on prior knowledge, simulation cost, hard exploration and data efficiency. Each curve in the graph represents the score of an algorithm across these six categories. A score of 1 indicates that the algorithm perform poorly in this dimension and is only applicable to limited scenarios, while a higer score means a large application scope and better performance. In particular, model-free RL methods need no simulation and have little dependence on priors, so it achieves high score in corresponding dimensions. Please note that within this context, the term *LightZero* refers to the specical algorithm that embodies the optimal combination of techniques and hyperparameter settings within our framework. Details about qualitative score rules can be found in Appendix D.

where $T$ is the temperature coefficient controlling the exploration degree. Finally, an action is sampled from this distribution for interaction with the environment or self-play. During the learning phase, MuZero perform end-to-end training with the following loss function, where $l^{\mathrm{p}}$, $l^{\mathrm{v}}$ and $l^{\mathrm{r}}$ are loss functions for policy, value and reward respectively, and the final term is weight decay.

$$l_t(\theta) = \sum_{k=0}^{K} l^{\mathrm{p}}(\pi_{t+k}, p_t^k) + l^{\mathrm{v}}(z_{t+k}, v_t^k) + l^{\mathrm{r}}(u_{t+k}, r_t^k) + c||\theta||^2$$

## 3 LightZero

In this section, we will first introduce the overview of LightZero, followed by a comprehensive analysis of challenges in various decision environments. Additionally, we propose a specific training pipeline design for a modular and scalable MCTS toolchain. We will conclude this section with two algorithm insights inspired by the decoupled design of LightZero.

### 3.1 Overview

As is shown in Figure 1, LightZero is the first benchmark that integrates almost all recent advances in the MCTS/MuZero sub-domain. Specifically, LightZero incorporates nine key algorithms derived from the original AlphaZero [29], establishing a standardized interface for training and deployment across diverse decision environments. Unlike the original versions of these derived algorithms, which focused on specific avenues of improvement, LightZero provides a unified viewpoint and interface. This unique feature enables exploration and comparison of all possible combinations of these techniques, offering an comprehensive baseline for reproducible and accessible research. The concrete experimental results are thoroughly described in Section 4 and Appendix B.

### 3.2 How to Evaluate A General MCTS Algorithm: 6 Environment Challenges

The algorithm extensions integrated in LightZero have greatly relaxed the constraints and broadened the applicability of MCTS-style methods. In the following part, we hope to delve deeper into the key

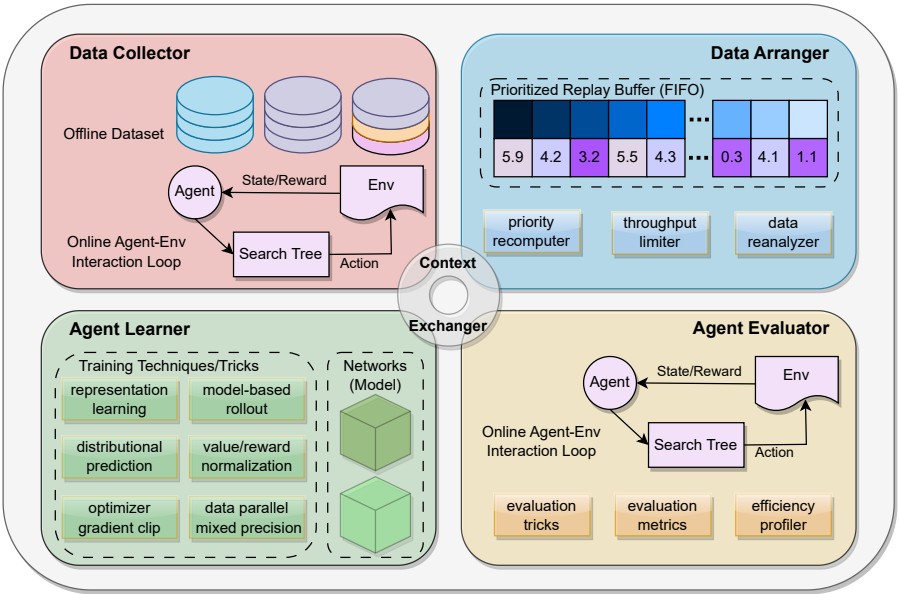

Figure 3: Four core sub-modules of the training pipeline in LightZero. *Context Exchanger* is responsible for transporting configurations, models and trajectories among different sub-modules.

issues in the design of general and efficient MCTS algorithms. In order to systematically complement this endeavor, we conducted an analysis of a set of classic and newly proposed RL environments to identify common characteristics. Based on this analysis, we have summarized six core challenging dimensions, which are presented in a radar plot depicted in Figure 2. Concretely, The intentions and goals of six types of envionmental capabilities are: *1) Multi-modal observation spaces* pose a challenge for agents as they must be able to extract different representation modalities (e.g., low-dimensional vectors, visual images, and complex relationships) while effectively fusing distinct embeddings. *2) Complex action space* necessitates the agent's proficiency in generating diverse decision signals, encompassing discrete action selection, continuous control, and hybrid structured action space. *3) Reliance on prior knowledge* is a major drawback of methods like AlphaZero. These approaches inherently require accessibility to a perfect simulator and specific rules of the environment. In contrast, MuZero and its derived methods address this limitation by learning an environment model to substitute the simulator and related priors. *4) Inherent stochasticity* presents a fundamental challenge in tree-search-based planning methods. The uncertainty of environment dynamics and partially observable state spaces both can lead to misalignment of planning trajectories, resulting in a large number of useless or conflicting search results. *5) Simulation cost* stands as the primary contributor to wall-time consumption for MCTS-style methods. At the same time, the algorithm performance will degrade a lot if the algorithm fails to visit all the necessary actions during the simulation process. *6) Hard exploration* represents a crucial challenge that is often overlooked. While search trees can enhance efficiency by reducing the scope of exploration, MCTS-style methods are susceptible to difficulties in environments with numerous non-terminating cases, such as mazes.

### 3.3 How to Simplify A General MCTS Algorithm: Decouple Pipeline into 4 Sub-Modules

The impressive performance of MCTS-style methods is often accompanied by a notable drawback: the complexity of implementations, which greatly restricts their applicability. In contrast to some classic model-free RL algorithms like DQN [32] and PPO [5], MCTS-style methods require multi-step simulations using search trees at each agent-environment interaction. Also, to improve the quality of training data, MuZero Unplugged [24] introduce a data reanalyze mechanism that uses the newly obtained model to compute improved training targets on old data. However, both of these techniques require multiple calls to simulators or neural networks, increasing the complexity across various aspects of the overall system, including code, distributed training, and communication topology.

Therefore, it is necessary to simplify the whole framework based on the integration of algorithms. Figure 3 presents a depiction of the complete pipeline of LightZero with four core sub-modules.

Firstly, LightZero offers support for both online and offline RL [33] training schemes. The distinction between them lies in the utilization of either an online interaction data collector or direct usage of an offline dataset. Secondly, LightZero restructures its components and organizes them into four main sub-modules, based on the principle of *high cohesion and low coupling*. **Data collector** is responsible for efficient action selection using policy network and search tree. It also contains various exploration strategies, data pre-processing and packaging operations. **Data arranger** plays a unique role in MCTS by effectively storing and preparing valuable data for training purposes. This sub-module involves the data reanalyze technique [24] to correct off-policy and even offline data. Furthermore, the modified priority sampling [34] ensures training mini-batches have both sufficient variety and high learning potential. To balance these tricks with efficiency, the throughput limiter controls the ratio of adding and sampling data to ensure optimal data utilization within a fixed communication bandwidth. **Agent learner** is responsible for training multiple networks. It can be enhanced through various optimization techniques, such as self-supervised representation learning [35, 36], model-based rollout [37, 38], distributional predicton [39] and normalization [40, 41]. These techniques contribute to the policy improvement and further enhance the overall performance of the agent. **Agent evaluator** periodically provides the diverse evaluation metrics [42] to monitor the training procedure and assess policy behaviour. It also integrates some inference-time tricks like beam search [43] to enhance test performance. We provide a detailed analysis of how these sub-modules are implemented in specific algorithms in Appendix F. Built upon these abstractions, LightZero serves as a valuable toolkit, enabling researchers and engineers to develop enhanced algorithms and optimize systems effectively. For example, the exploration strategies and ensuring the alignment of a learned model in MCTS is crucial, and this will be discussed in the subsequent sub-section. In addition, exploring parallel schemes for multiple vectorized environments and search trees can be an insightful topic for machine learning system. The associated dataflow and overhead analysis will be presented in the Appendix E.

### 3.4 How to Improve A General MCTS Algorithm: 2 Examples

In this section, we present two algorithm improvement examples inspired by LightZero. The below dimensions pose necessary challenges in designing a comprehensive MCTS solver. LightZero addresses these challenges through various improvements, resulting in superior performance compared to individual algorithm variants across different domains (Section 4 and 5).

**Intrinsic Exploration** While tree-search-based methods perform well in board games with only eventual reward, they may encounter challenges or perform poorly in other environments with sparse rewards, such as *MiniGrid* [44]. One crucial distinction between these two problems is that in the former, the search tree can always reach several deterministic final states, whereas in the latter, it may encounter various non-termination states due to the limitation of maximum episode length. To address this issue, LightZero incorporates the idea of intrinsic reward methods [28] and implement it efficienctly within MuZero's learned models. Further details can be found in Section 5.1.

**Alignment in Environment Model Learning** MuZero employs a representation network to generate latent states and a dynamics network to predict next latent states. However, there is no explicit supervision guiding the desired properties of the latent space. Traditional self-supervised representation learning methods often fail to align these proxy tasks with RL objectives. The difference of rollouts between the perfect simulator and the learned model is also a problems that can not be ignored. Further exploration of misalignments across different environments are discussed in Section 5.2.

## 4   Experiments

In Section 4.1, we initially present some representative experiments of LightZero, with detailed experimental settings and more comprehensive results outlined in the Appendix B. Subsequently, in Section 4.2, we delve into key observations and reflections based on these benchmark results, introducing some critical insights. Particularly regarding the exploration and the alignment issues of environment model learning, we conduct an in-depth experimental analysis in Section 5.

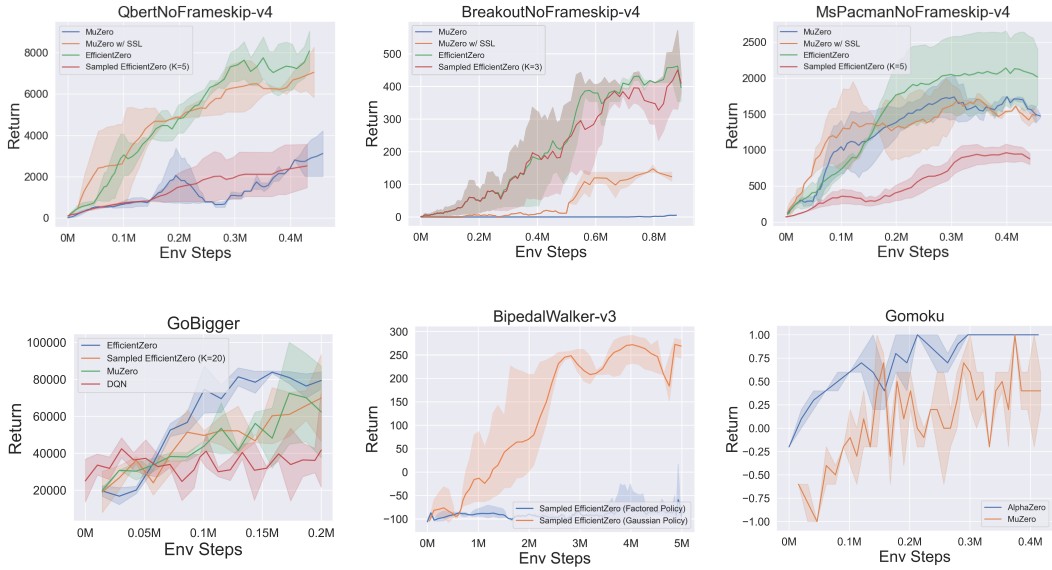

Figure 4: Comparisons of mean episode return for algorithm variants in LightZero across diverse environments: *Atari* with discrete action and partial-observable state (*Qbert*, *Breakout*, *MsPacman*), *GoBigger* [23] with complex observation and multi-agent cooperation, continuous control with environment stochasticity (*Bipedalwalker*), and *Gomoku* with varying accessibility to simulator.

## 4.1 Benchmark Results

To benchmark the difference among distinct algorithms and the capability of LightZero as a general decision solver, we conduct an extensive comparisons across a diverse range of RL environments. The algorithm variants list contains AlphaZero [29], MuZero [14], EfficientZero [17], Sampled MuZero [15], Stochastic MuZero [16], Gumbel MuZero [18] and other improved versions of LightZero. For each scenario, we evaluate all the possible variants on corresponding environments. In Figure 4, we show some selected results as examples. For detailed settings, metrics, comprehensive benchmark results and related analysis, please refer to Appendix B.

## 4.2 Key Observations and Insights

Building on the unified design of LightZero and the benchmark results, we have derived some key insights about the strengths and weaknesses of each algorithm, providing a comprehensive understanding of these algorithms' performance and potential applications.

**O1**: In board game environments, AlphaZero's sample efficiency greatly exceeds that of MuZero. This suggests that employing AlphaZero directly is advantageous when an environment simulator is available; however, MuZero can still achieve satisfactory results even in the absence of a simulator.

**O2**: Self-supervised loss substantially enhances performance in most Atari environments with image inputs. Figure 7 demonstrates that MuZero with SSL performs similarly to MuZero in *MsPacman*, while outperforming it in the other five environments. This result highlights the importance of SSL for aligning the model and accelerating the learning process in image input environments.

**O3**: Predicting *value_prefix* instead of reward does not guarantee performance enhancement. For example, in Figure 7, EfficientZero outperforms MuZero with SSL only in the *MsPacman* and *Breakout* environments, while showing similar performance in the other environments. In certain specific scenarios, such as the sparse reward environments depicted in Figure 12, EfficientZero's performance is significantly inferior to that of MuZero with SSL. Therefore, we should prudently decide whether to predict *value_prefix*, taking into account the attributes of the environment.

**O4**: MuZero with SSL and EfficientZero demonstrate similar performance across most *Atari* environments and in complex structured observation settings, such as *GoBigger*. This observation suggests

that environments with complex structured observations can benefit from representation learning and contrastive learning techniques [35] to enhance sample efficiency and robustness.

**O5**: In discrete action spaces, Sampled EfficientZero's performance is correlated with action space dimensions. For instance, Sampled EfficientZero performs on par with EfficientZero in *Breakout* (action space dimension of 4), but its performance decreases in *MsPacman* (dimension of 9).

**O6**: Sampled EfficientZero with *Gaussian policy representation* is more scalable in continuous action spaces. The Gaussian version performs well in traditional continuous control and *MuJoCo* environments, while *factored discretization* [45] is limited to low-dimensional actions.

**O7**: Gumbel MuZero achieves notably better performance than MuZero when the number of simulations is limited, which exhibits its potential in designing low time-cost MCTS agent.

**O8**: In environments with stochastic state transitions or partial observable states (such as Atari without stacked frames), Stochastic MuZero can obtain slightly better performance than MuZero.

**O9**: The self-supervised loss proposed in [17], sampling-related techniques in Sampled MuZero [45], computational improvements in Gumbel MuZero [18] for utilizing MCTS searched information, and environment stochasticity modeling in Stochastic MuZero [16] are orthogonal to each other, exhibiting minimal interference. LightZero is exploring and developing ways to seamlessly integrate these characteristics to design a universal decision-making algorithm.

## 5 Two Algorithm Case Studies for LightZero

### 5.1 Exploration Strategies in MCTS

**Motivation** Finding the optimal trade-off between exploration and exploitation is a fundamental challenge in RL. It is well-known that MCTS can reduce the policy search space and facilitate exploration. However, there exists limited research on the performance of MCTS algorithms in hard-exploration environments. Based on the above benchmark results, we conduct a detailed analysis of the algorithm behaviours between challenging sparse reward environments and board games, as well as insights behind the selection of exploration mechanisms in this section and Appendix C.1.

**Settings** We performed experiments in *MiniGrid* environment, mainly on the KeyCorridorS3R3 and FourRooms scenarios. Expanding upon the naive setting (handcrafted temperature decay), we conducted a comprehensive investigation of six distinct exploration strategies in LightZero. A detailed description of each exploration mechanism is provided in Appendix C.1.

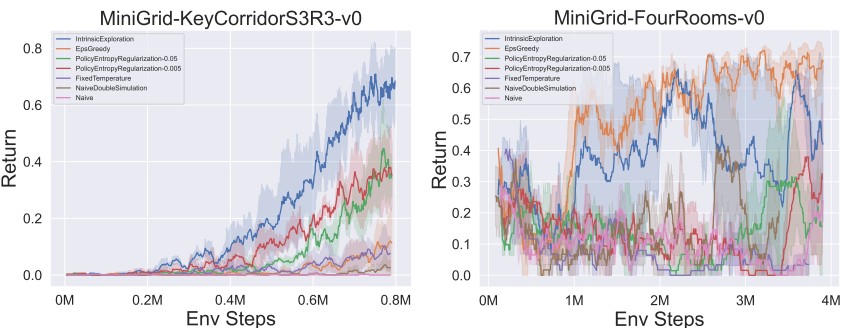

Figure 5: Performance of various MCTS exploration mechanisms in *MiniGrid* environment (*Return* during the collection phase). Under the naive setting, the agent fails due to inadequate exploration. Merely increasing search budgets with the *NaiveDoubleSimulation* approach does not yield any significant improvement. *EpsGreedy*, *FixedTemperature* and *PolicyEntropyRegularizatio-x* display higher variance as they cannot guarantee enough exploration. *IntrinsicExploration* effectively explores the state space by leveraging curiosity mechanisms, resulting in the highest sample efficiency.

**Analysis** Figure 5 indicate that simply increasing search budgets does not yield improved performance in challenging exploration environments. Instead, implementing a larger temperature and

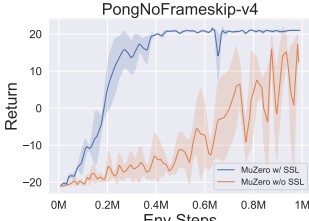
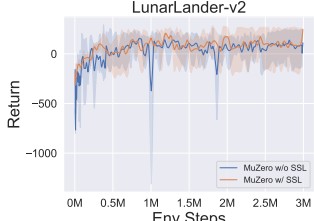
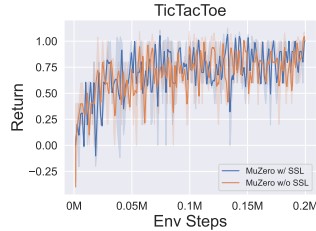

Figure 6: Impact of self-supervised consistency loss across different environments with various types of observations. From left to right, performance comparisons involve standard image input, compact vector input, and unique board image input, considering cases with and without consistency loss. Experiments show that the consistency loss proves to be critical only for standard image input.

incorporating policy entropy bonus can enhance action diversity during data collection, albeit at the cost of increased variance. However, theoretically, they cannot guarantee sufficient exploration, often resulting in mediocre performance and a higher likelihood of falling into local optima due to policy collapse. Epsilon-greedy exploration ensures a small probability of uniform sampling, which aids in exploring areas with potentially high returns. *EpsGreedy* has varying effects in different environments in early stages, but theoretically, due to its ability to ensure sufficient exploration, it may achieve good results in the long run. A more effective strategy involves curiosity-driven techniques, such as RND [27], which assigns higher intrinsic rewards to novel state-action pairs, bolstering the efficiency of exploration. The performance of the *IntrinsicExploration* method supports this assertion, and it can be integrated into MuZero with minimal overhead (Appendix C.1.3).

## 5.2   Alignment in Environment Model Learning

**Motivation** Aligned and scalable [25] environment models are vital for MuZero-style algorithms, with factors such as model structure, objective functions, and optimization techniques contributing to their success. The consistency loss proposed in [17] could serve as an approach for aligning the latent state generated by the dynamics model with the state obtained from the observation. In this section, we investigate the impact of consistency loss on learning dynamic models and final performance in environments with diverse observations (vector, standard images, special checkerboard images).

**Settings** To study the impact of the consistency loss on various types of observation data, we employ the MuZero algorithm as our baseline. To ensure the reliability of our experimental results, we maintain the same configurations across other settings, with additional experimental details provided in Appendix C.2. In the experiments, we use *Pong* as the environment for image input, *LunarLander* for continuous vector input, and *TicTacToe* for special image input (checkerboard) environments.

**Analysis** In Figure 6, consistency loss is critical for standard image input. Removing the consistency loss results in significant decline in performance, indicating the challenge of learning a dynamic model for high-dimensional inputs. For vector input environments like *LunarLander*, consistency loss provides a minor advantage, suggesting that learning a dynamic model is relatively easier on the compact vector observations. In special two dimension input environments like *TicTacToe*, the consistency loss remains large, highlighting the difficulty of achieving consistency between latent state outputs. Additionally, adding consistency loss with inappropriate hyper-parameters may lead to non-convergence (Appendix C.2). In conclusion, our experiments demonstrate that the effectiveness of the consistency loss depends on the special observation attributes. For board games, a future research direction involves investigating suitable loss functions to ensure alignment during training.

## 6   Related Work

**Sequential Decision-making Problems** In the domain of sequential decision-making problems, intelligent agents aim to make optimal decisions over time, taking into account observed states and prior actions [46]. However, these problems are often compounded by the presence of continuous action spaces, dynamic transitions, and exploration difficulties. To address such problems, model-free RL methods [5, 7, 32] focus on learning expected state rewards, optimizing actions, or combining

both strategies to achieve optimal policy learning. Conversely, model-based RL [25] incorporates the environment's transition into its optimization objective, aiming to maximize the expected return on trajectory distribution. MCTS [19] is a modeling approach derived from search planning algorithms such as minimax [47] and alpha-beta pruning [48]. Unlike these algorithms, which recursively search decision paths and evaluate their returns, MCTS employs a heuristic search on prior-guided simulations, effectively addressing excessive search consumption in complex decision spaces.

**MCTS Algorithms and Toolkits** Despite the impressive performance and efficiency of the MCTS+RL approach, constructing the training system and dealing with intricate algorithmic details pose significant challenges when applying these algorithms to diverse decision intelligence domains. Recent research has made progress in this direction. MuZero Unplugged [24] introduced the Reanalyze technique, a simple and efficient enhancement that achieves good performance both online and offline. ROSMO [49] investigated potential issues with MuZero in offline RL and suggested a regularized one-step lookahead approach. The lack of comprehensive open-source implementations of various algorithms remains a challenge within the research community. For example, Sampled MuZero [15] lacks a public implementation. AlphaZero-General [50] and MuZero-General [51] each support only a single algorithm, and neither offers distributed implementations. Although EfficientZero [17], does support multi-GPU implementation, it is limited to the single algorithms. KataGo [52], while primarily focused on the AlphaZero and Go game, requires substantial computational resources during training, potentially posing hardware barriers for ordinary users. As a result, the research community continues to seek more efficient and enhanced open-source tools.

**Standardization and Reproducibility** In the realm of Deep RL, the quest for standardizing algorithm coupled with the creation of unified benchmarks has ascended as focal points of growing significance. [53] emphasize the critical necessity of not only replicating existing work but also accurately assessing the advancements brought about by new methodologies. However, the process of reproducing extant Deep RL methods is far from straightforward, largely due to the non-determinism inherent in environments and the variability innate to the methods themselves, which can render reported results challenging to interpret. [54] proposed a set of metrics for quantitatively measuring the reliability of RL algorithms. These metrics, focusing on variability and risk both during and after training, are intended to equip researchers and production users with tools to evaluate and enhance the reliability of RL algorithms. In [55], a large-scale empirical study was undertaken to identify the factors that significantly influence the performance of on-policy RL algorithms within continuous control tasks. The insights garnered from this research offer valuable, practical suggestions for the training of on-policy RL algorithms. Despite these advancements, there remains a noticeable dearth of work specifically investigating benchmarks and the details of reproducing studies in the domain of MCTS.

## 7 Conclusion

In this paper, we introduce LightZero, the first unified algorithm benchmark to modularly integrates various MCTS-style RL methods, systematically analyze and address the challenges and opportunities for deploying MCTS as a general and efficient decision solver. Through the incorporation of decoupled system design and novel algorithm insights, we conduct detailed benchmarks and demonsrate the potential of LightZero as scalable and efficient decision-making problem toolchains for the research community. Besides, based on this benchmark and related case studies, We also discuss existing limitations and valuable topics for future work in Appendix I.

## 8 Acknowledgements

This project is funded in part by National Key R/D Program of China Project 2022ZD0161100, by the Centre for Perceptual and Interactive Intelligence (CPII) Ltd under the Innovation and Technology Commission (ITC)'s InnoHK, by General Research Fund of Hong Kong RGC Project 14204021. Hongsheng Li is a PI of CPII under the InnoHK. We thank several members of the SenseTime and Shanghai AI Laboratory for their help, support and feedback on this paper and related codebase. We especially thank Shenghan Zhang for informative and inspiring discussions at the beginning of this project. We are grateful to the assistance of Qingzi Zhu for many cute visual materials of LightZero.

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

# A  Environment

| Environment Attributes | Possible Categories | Environment Instances |
|---|---|---|
| Observation Space | • Vector Observation Space | • *Classical Control* [56], *Bsuite* [57], *Box2D* [58], *DMControl* [59], *Gym-Hybrid* [60] |
| | • Image Observation Space | • *Board Games, Atari* [30], *Bsuite, MiniGrid* [44], *ProcGen* [61], *2048* |
| | • Structured Observation Space | • *MetaDrive* [62], *GoBigger*[23] |
| Action Space | • Discrete Action Space | • *Board Games, Atari, Bsuite, MiniGrid, ProcGen, 2048* [63] |
| | • Continuous Action Space | • *MuJoCo* [64], *Classic Control, Box2D, DMControl, MetaDrive* |
| | • Hybrid Action Space | • *Gym-Hybrid, GoBigger* |
| Reward Space | • Sparse Reward | • *MiniGrid, Board Games, Atari, ProcGen, Bsuite* |
| | • Normal Dense Reward | • *Classical Control, Bsuite, Box2D, DMControl, Gym-Hybrid, Atari, ProcGen, 2048* |
| | • Multi-Objective Dense Reward | • *MetaDrive, GoBigger* |
| Transition Function | • Stochastic | • *2048, Atari, Bsuite* |
| | • Back-traceable | • *Board Games, 2048, MiniGrid* |

Table 1: **Overview of various attributes associated with decision-making environments, their respective potential categories crucial for the design of MCTS-style methods, and specific instances of environments that exemplify each category**. Please note that each environment can encompass a series of different sub-environments (e.g. *Atari* typically consists of 57 distinct video games), thus some environments can simultaneously belong to multiple categories. While most categories are self-explanatory, there are two special cases worth noting: *Multi-Objective Dense Reward* indicates that there are multiple, distinct rewards to be balanced for varying decision targets (e.g. balancing route optimization, speed, and stability in autonomous driving tasks). *Back-traceable* is a critical property for *AlphaZero-like* methods that requires a perfect simulator to return to the original state after each simulation. While *Board Games* can easily store and restore old states, most control tasks and video games are practically impossible to revert to their previous states.

In this section, we introduce various types of RL environments integrated in LightZero and their respective characteristics. These environments encompass a wide range, including *Board Games, Classical Control, Atari, MuJoCo*, the sparse reward example environment (*MiniGrid*), the structured action space example environment (*GoBigger*). We categorize these environments based on different attributes in Table 1, followed by a detailed description of each environment.

**Board Games** This types of environment includes *TicTacToe*, *Connect4*, *Gomoku*, *Chess*, *Go*, where uniquely marked boards and explicitly defined rules for placement, movement, positioning, and attacking are employed to achieve the game's ultimate objective. These environments feature a variable discrete action space, allowing only one player's piece per board position. In practice, algorithms utilize action mask to indicate reasonable actions.

**Classic Control** In the reinforcement learning domain, classical control environments like *Cartpole* and *Pendulum* are built upon physical principles and provide explicit action and state spaces. They are commonly used as benchmark environments to evaluate the initial performance of reinforcement learning algorithms in discrete or continuous action spaces.

**Box2D** All of these environments in this suite feature toy games centered around physics control (e.g., *LunarLander* and *BipedalWalker*), utilizing Box2D-based physics and PyGame-based rendering.

Their straightforward simulation mechanics and meaningful visual outputs have made them popular benchmarks for RL beginners. Moreover, they are suitable for visualizing the tree search process.

**Atari** This category includes sub-environments like *Pong, Qbert, Ms.Pacman, Breakout, UpNDown*, and *Seaquest*. In these environments, agents control game characters and perform tasks based on pixel input, such as hitting bricks in *Breakout*. With their high-dimensional visual input and discrete action space features, Atari environments are widely used to evaluate the capability of reinforcement learning algorithms in handling visual inputs and discrete control problems.

**MuJoCo** This category includes sub-environments such as *Hopper-v3, HalfCheetah-v3, Walker2d-v3*, and *Humanoid-v3*. *MuJoCo* is a precise and efficient physics simulation engine capable of simulating intricate digital mechanical systems, such as bipedal robots and robotic arms. These environments serve as testbeds for assessing the performance of reinforcement learning in control tasks like robot balancing or robotic arm object grasping. *MuJoCo* environments are characterized by continuous state and action spaces, and real-world physical laws, commonly used to evaluate the ability of reinforcement learning algorithms to handle continuous control and high-dimensional input problems.

**MiniGrid** In addition to the difficulty levels of action and observation spaces, the sparsity or density of reward spaces is another major consideration. For example, MiniGrid series environments, like *MiniGrid-Empty-8x8-v0* and *MiniGrid-FourRooms-v0*, provide simple and scalable sparse reward environments. In these environments, agents navigate in a grid world and complete various tasks. Sparse reward environments pose challenges to the exploration capabilities of reinforcement learning algorithms, as agents receive limited useful feedback most of the time.

**GoBigger** *GoBigger* is a multi-agent reinforcement learning environment that emphasizes cooperation and competition. It features structured observation spaces where each agent is represented by one or more clone balls. The goal of the agents is to collide and merge with other balls within a limited time, thereby increasing their size. The observation space includes all unit information within the agent's local view, and rewards depend on the size difference between consecutive time steps. This environment offers diverse sub-environments for different tasks, such as *t2p2, t3p2, t4p3*, etc. Here, the number following $t$ represents the number of teams, and the number following $p$ indicates the number of agents per team. In our experiments, we set $t = p = 2$.

**2048** *2048* is a numerical game in which players need to merge adjacent blocks with the same number in a 4x4 grid by swiping up, down, left, or right. The randomness of the game lies in the fact that after each swipe, a block with a value of 2 or 4 is randomly generated. Theoretically, there are up to 32 possibilities after each move, which poses challenges for model-based RL algorithms. For algorithms in the MCTS family, solving such problems requires modeling the randomness of the environment and considering various possibilities during the search process.

## B  Details about Main Experiment

### B.1  Benchmark Settings

**Environments** In Table 1, we provide a categorization of common sequential decision-making environments, each presenting distinct challenges. These environments encompass the majority of the cases previously discussed in Section 3.2. We have undertaken a thorough and impartial evaluation of the algorithms incorporated into LightZero, across these varied environments.

**Algorithms** In our main benchmark experiments, we assessed a series of algorithm variants, including *AlphaZero, MuZero, MuZero w/ SSL* (MuZero with Self-Supervised Learning Consistency Loss), *EfficientZero, Sampled EfficientZero, Stochastic MuZero*, and *Gumbel MuZero*.

Note that in this context, *MuZero w/ SSL* represents the original *MuZero* algorithm augmented with the self-supervised loss proposed in [17]. *EfficientZero* refers to the *MuZero* algorithm enhanced with the *self-supervised loss* and *value_prefix* from [17]. We decided not to open the third improvement discussed in the [17], as it was shown to have limited performance benefits and be highly time-consuming. Consequently, it was omitted from our primary benchmark experiments. *Sampled*

*EfficientZero* is based on the previously described *EfficientZero*, incorporating sampling-related modifications from [45]. This enables the algorithm to simultaneously handle environments with discrete and continuous action spaces.

**Metrics** In all our experimental evaluations, the primary metric employed to gauge performance was the mean *Return* calculated over the evaluated episodes for each algorithm. Each algorithm was independently executed five times with unique seeds to ensure the integrity and reliability of our results. In the results graphs, the shaded regions illustrate the standard deviation in different runs, indicating the variability or spread of the runs under different seeds. On the other hand, the solid lines represent the mean, providing the average performance across the different runs.

The horizontal axis, labeled as *Env Steps* (environment steps), represents the number of steps taken in the environment during each run. The vertical axis represents the average *Return* over the evaluated 20 episodes, serving as a measure of the algorithm's effectiveness in each run.

## B.2  Main Benchmark

In this section, we provide an comprehensive benchmark results for LightZero. This encompasses baseline performances in discrete action spaces detailed in Section B.2.1, and continuous action spaces in Section B.2.2. In addition, we include comparative analyses of *Gumbel MuZero* and *Stochastic MuZero*, delineated in Sections B.2.3 and B.2.4 respectively. Furthermore, we extend our investigation to present the baseline evaluations in sparse reward scenarios like *MiniGrid* (Section B.2.5), and in multi-agent environments exemplified by *GoBigger* (Section B.2.6).

For a thorough examination and in-depth discussion of these benchmark results, we encourage readers to refer to Section 4.2, where we provide key observations and insights derived from our experiments.

### B.2.1  Discrete Decision Benchmark

In Figure 7, we present a comparative performance evaluation of four key algorithms—*MuZero, MuZero with SSL, EfficientZero*, and *Sampled EfficientZero*—across a selection of six representative environments from the classic *Atari* image input domain.

Figure 8 showcases a performance comparison of *AlphaZero* and *MuZero* on two representative board games—*Connect4* and *Gomoku (board_size=6)*. What stands out is that while AlphaZero's sample efficiency significantly surpasses that of MuZero, the latter still manages to produce satisfactory results, even in the absence of a simulator.

### B.2.2  Continuous Control Benchmark

In the upper of Figure 9, we present the performance of *Sampled EfficientZero* with different policy modeling approaches on the classical continuous benchmark environment, *Pendulum-v1* and *LunarlanderContinous-v2*. In the bottom of Figure 9, we present the performance of Sampled EfficientZero with different policy modeling approaches on the conventional continuous benchmark environment, *MuJoCo*, examplifed by *Hopper-v3* and *Walker2d-v3*.

The dimensions of the action space for the four aforementioned environments are 1, 2, 3, and 6, respectively, with the number of discrete bins set to 11, 7, and 5. Consequently, the final dimensions of the factored categorical distributions [65] stand at $11^1 = 11, 7^2 = 49, 5^3 = 125$, and $5^6 = 15625$, in the same order. Notably, the dimensionality of the discretized action space undergoes an exponential increase with the dimension of the original continuous action space. Under conditions of a fixed simulation cost, specifically with the number of sampled actions $K$ set to 20, and the number of simulations per search $sim$ set to 50, we witness a steady decline in the performance of the *factored policy representation* as the dimensionality increases. Conversely, the *Gaussian policy representation* maintains a relatively stable performance, demonstrating its robustness in higher-dimensional spaces.

However, even the *Gaussian policy representation* does not achieve the performance of model-free methods in *MuJoCo*. We speculate that this is due to the unique reward mechanism of the

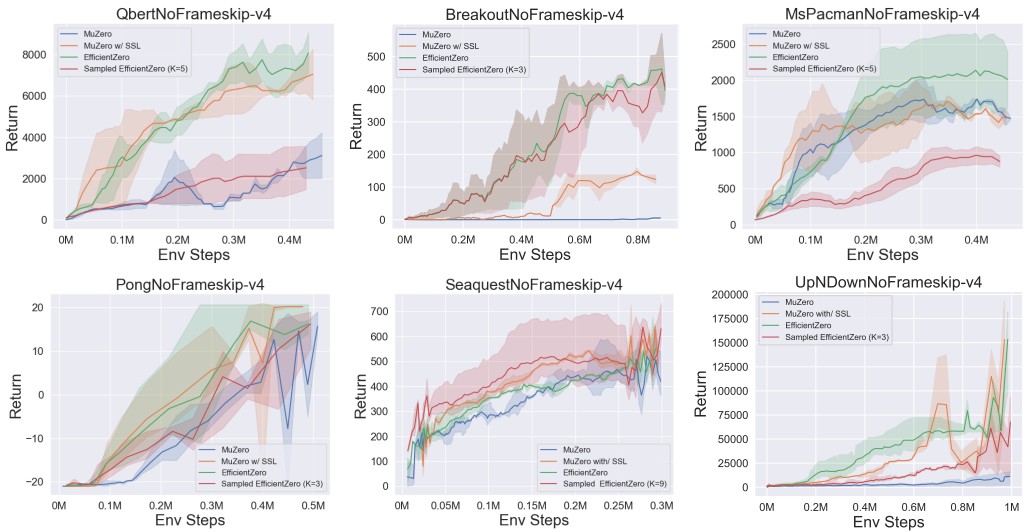

Figure 7: Performance comparison of algorithms integrated in LightZero on six representative image-based *Atari* environments. The horizontal axis represents *Env Steps* (environment steps), while the vertical axis indicates the average *Return* over 20 assessed episodes. In this context, *MuZero w/ SSL* denotes the original MuZero algorithm augmented with *self-supervised loss*. *EfficientZero* refers to the *MuZero* algorithm enhanced with *self-supervised loss* and *value_prefix*. *Sampled EfficientZero* builds upon *EfficientZero* with additional modifications related to sampling.

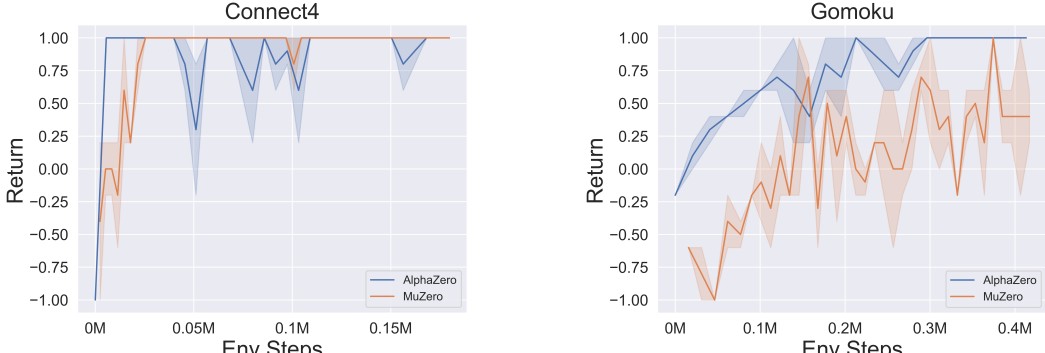

Figure 8: Performance comparison of *AlphaZero* and *MuZero* on *Connect4* and *Gomoku* *(board_size=6)*. AlphaZero exhibits significantly higher sample efficiency compared to MuZero. This suggests that deploying AlphaZero directly could be advantageous when an environment simulator is readily available. However, even in the absence of a simulator, MuZero can still yield comparable results, which underscores its broad applicability.

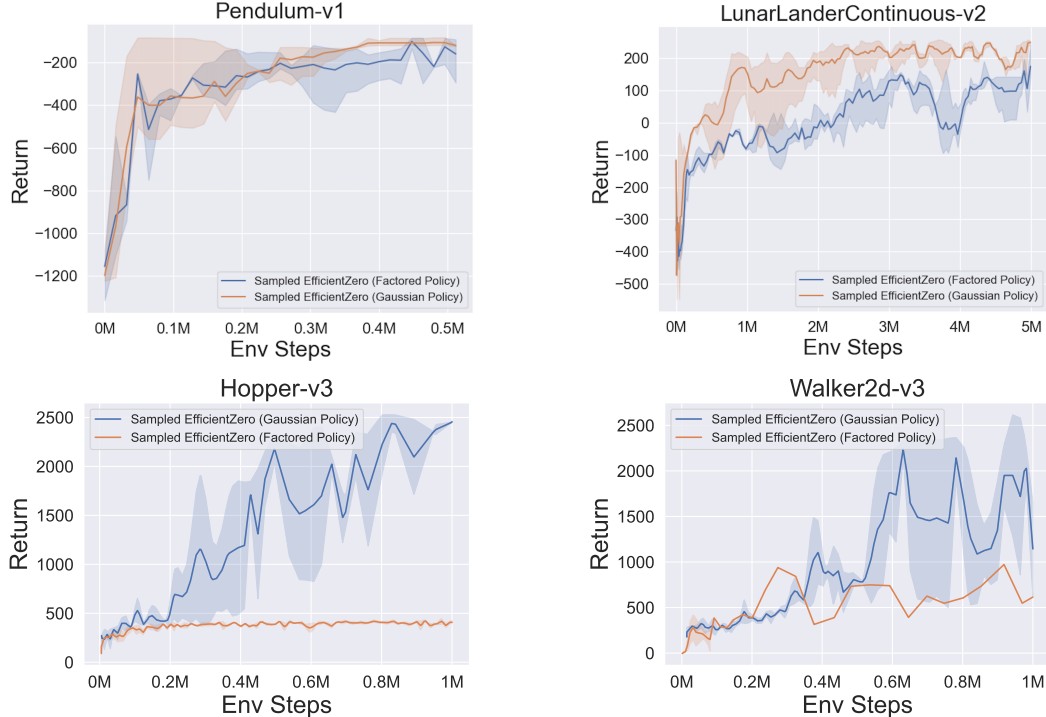

Figure 9: **Upper**: Performance comparison of *Sampled EfficientZero* utilizing different policy modeling techniques in continuous action space environments. **Bottom**: Performance comparison of *Sampled EfficientZero* using various policy modeling methods in *MuJoCo* continuous action space environments. Performance of the factored policy representation declines gradually with increasing action space size, while the Gaussian policy representation exhibits more stable performance.

*MuJoCo* environment, where the optimal policy actions are likely extreme values of $-1$ or $1$ in most times. Sampling from a Gaussian distribution makes it challenging to obtain these extreme actions, which leads to suboptimal performance. To enhance performance in *MuJoCo* environments, a straightforward idea for future work is to augment extreme actions during the sampling process.

### B.2.3 Gumbel MuZero Benchmark

In Figure 10, we present the performance comparison of *Gumbel MuZero* and *MuZero* under different simulation costs (i.e. the number of simulations in one search) on *Gomoku* (*board_size=6*), *Lunarlander-v2*, *PongNoFrameskip-v4* and *Ms.PacmanNoFrameskip-v4*. Original Monte Carlo Tree Search methods have to visit almost all the possible nodes to ensure stable optimization. The demands of high simulation counts have become the main time-consuming source of MCTS-style methods. In our benchmark, we validate that *Gumbel MuZero* can achieve notably better performance than *MuZero* when the number of simulations is limited. Besides, we don't tune the balance weight of the *completedQ* proposed in Gumbel MuZero, it may siginicantly influence the final performance in some environments, which maybe the reason why Gumbel MuZero shows lower episode return in the high simulation times case.

### B.2.4 Stochastic MuZero Benchmark

In Figure 11, we present a comparative performance analysis between *Stochastic MuZero* and *MuZero*, conducted within *2048* environments that exhibit intrinsic randomness. Specifically, our attention is focused on the environments typified by standard *2048* settings, where *num_chances=2* (potential random tiles being {2, 4}), and a variant environment with increased stochasticity, *num_chances=5* (potential random tiles are {2, 4, 8, 16, 32}). *Stochastic MuZero* exhibits a distinct performance advan-

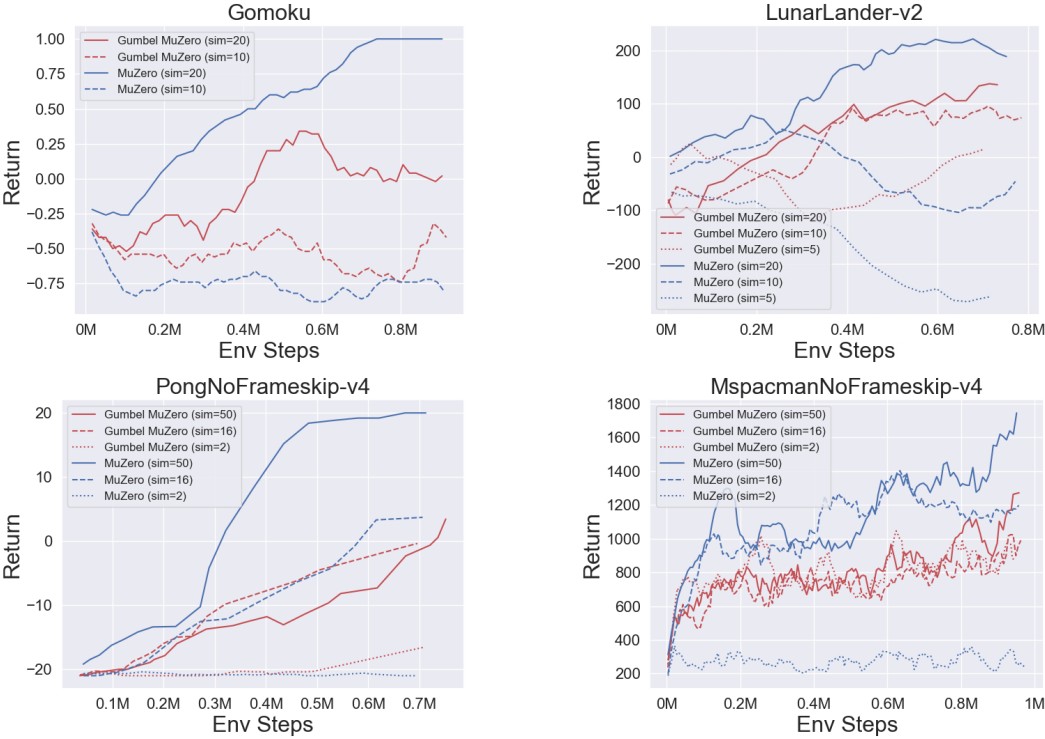

Figure 10: Performance comparison of *Gumbel MuZero* and *MuZero* under varying simulation costs. *Gumbel MuZero* demonstrates significantly better performance than *MuZero* when the number of simulations is limited, showcasing its potential for designing low time-cost MCTS agents. For *Gomoku (board_size=6)*, we evaluate *sim*={20, 10}. For *LunarLander-v2*, we assess *sim*={20, 10, 5}. For *Atari* Games, we examine *sim*={50, 16, 2}.

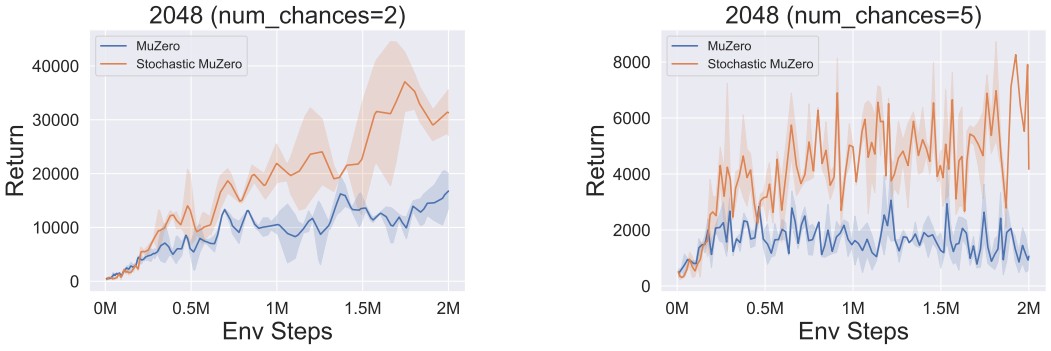

Figure 11: Performance comparison between *Stochastic MuZero* and *MuZero* in *2048* environments with varying levels of chance (*num_chances*=2 and 5). In environments characterized by stochastic transition dynamics, *Stochastic MuZero* marginally surpasses *MuZero*. However, as the level of stochasticity escalates, the performance of *Stochastic MuZero* begins to face limitations.

tage over *MuZero* when the environment's transition encapsulates elements of stochasticity. However, as the stochasticity escalates, the performance of *Stochastic MuZero* begins to face limitations. This implies that intricate randomness remains a substantial challenge for MCTS algorithms.

### B.2.5    MiniGrid Benchmark

In Figure 12, we present a performance comparison using the example environment *MiniGrid-KeyCorridorS3R3-v0* from the MiniGrid suite. The figure on the left demonstrates the application of

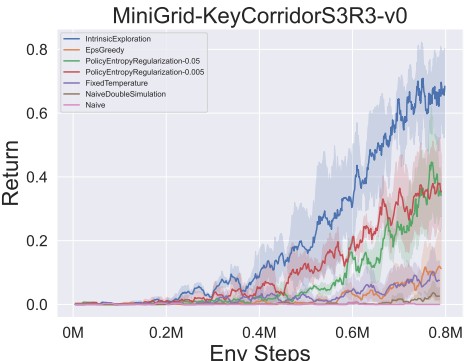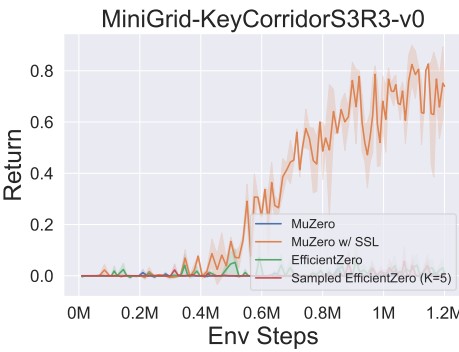

Figure 12: **Left**: Performance comparison of various exploration strategies applied to the *MiniGrid-KeyCorridorS3R3-v0* environment (*Return* during the collection phase). The *IntrinsicExploration* strategy, which leverages curiosity mechanisms to explore the state space, demonstrates superior sample efficiency. **Right**: Performance comparison of algorithms implemented in LightZero on *MiniGrid-KeyCorridorS3R3-v0*. In environments characterized by high-dimensional vector observations and sparse rewards, the *self-supervised learning* loss assists model alignment. However, predicting the *value_prefix* can pose challenges and potentially impede learning.

various exploration strategies on this environment, utilizing MuZero with Self-Supervised Learning (SSL). The right figure shows four algorithms—*MuZero, MuZero with SSL, EfficientZero*, and *Sampled EfficientZero* on it. From the experimental results, it can be observed that only MuZero with SSL achieved near-optimal performance, while the other three algorithms did not show significant progress. This suggests that in high-dimensional vector input environments with sparse rewards, *self-supervised learning* loss is crucial for aligning model learning. However, predicting the *value_prefix* may be more challenging than predicting the reward, which could, in turn, hinder the learning process.

### B.2.6 Multi Agent Benchmark

Compared to single-agent environments, multi-agent environments confront more complex challenges, including, but are not limited to, joint state-action spaces, multi-dimensional optimization objectives, non-stationarity and credit assignment issues [66]. To tackle these challenges, our preliminary experiments in multi-agent environments mainly adopted the independent learning paradigm [67]. In this paradigm, we regard other agents as part of the environment, with each agent making decisions independently, and all agents sharing a common policy-value network. During the Monte Carlo Tree Search process, each agent also conducts searches independently.

To investigate the impact of the number of agents on algorithm performance, we conducted experiments in the GoBigger environment [23] using our custom *T2P2* and *T2P3* scenarios (all other environment parameters remained the same except for the number of agents (*P*)). As demonstrated in Figure 13, in both two scenarios, our algorithm can achieve stable convergence in confrontations with built-in bots, and its sample efficiency has improved about six-fold compared to non-MCTS methods. For comparison, in the *T2P2* scenario, the *MuZero* algorithm requires approximately 400K *Env Steps* to reach a return of 150K, while algorithms such as MAPPO [68] necessitate about 3M *Env Steps* to achieve the same level of performance according to the paper [23]. This evidence proves that, aside from increased time expenditure, the performance of the MCTS algorithm is not significantly impacted when the number of agents is not very large. These initial attempts underscore the vast potential of sample-efficient MCTS-like algorithms in multi-agent environments.

In the future, we aim to provide baseline results of various algorithms integrated into LightZero on more multi-agent baseline environments (such as PettingZoo [69], MAMuJoCo [70]). Additionally, we plan to delve deeper into how to leverage the unique characteristics of multi-agent environments and multi-agent reinforcement learning (MARL) techniques such as Centralized Training Decentral-

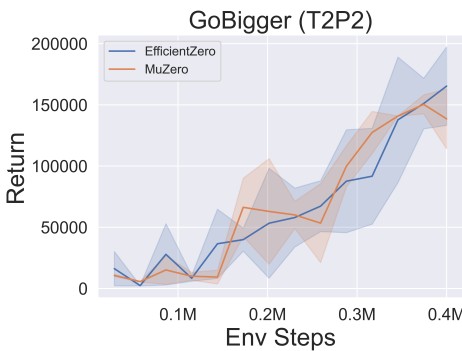
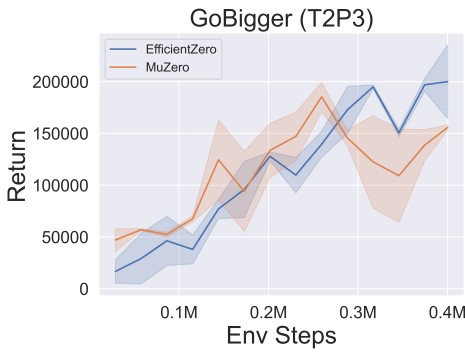

Figure 13: Performance comparison between MuZero and EfficientZero, both trained in the independent learning paradigm on the representative multi-agent environment GoBigger in *T2P2* and *T2P3* scenarios. Both algorithms demonstrate the ability to achieve stable convergence when pitted against built-in bots, and their sample efficiency showcases about a six-fold improvement over other non-MCTS methods [23].

ized Execution (CTDE) paradigm [66] [71] to conduct efficient MCTS searches, with the goal of designing superior cooperative and adversarial agents.

# C Details about Two Case Studies

## C.1 Exploration Mechanism in MCTS

### C.1.1 Details about Motivation

Striking the right balance between exploration and exploitation [26] [27] [28] is a fundamental challenge in reinforcement learning. For the MCTS algorithms integrated into LightZero, determining the optimal balance is also crucial. However, there has been limited research on the performance of MCTS algorithms in sparse reward environments. Recently, [72] investigated the combination of adversarial imitation learning and MCTS in the DeepMind Control Suite [59]; however, they did not delve into performance in sparse reward environments like *MiniGrid* [44].

Our preliminary experiments revealed that in some simple sparse reward environments in *MiniGrid*, MuZero struggles to make substantial progress within 1M steps. We observed that while board games also exhibit sparse rewards, **each game has a clear win/draw/loss outcome**, providing a supervisory signal for assessing the quality of the game state. In contrast, in other sparse reward environment like *MiniGrid*, **non-zero rewards are only obtained upon reaching the goal, with all other states yielding zero rewards**. This makes it challenging to collect trajectories with supervisory signals. As searching in some intermediate states of *MiniGrid* may be of limited value due to zero rewards, we hypothesize that increasing the number of simulations may not significantly improve performance.

In Table 2, we provide a comprehensive comparison of exploration strategies employed in MCTS algorithms as part of the LightZero framework. In this study, our main emphasis is on the general exploration strategies applied to decision-making algorithms. As for the unique exploration parameters specific to MCTS, previous research has predominantly used default values, which are likely near-optimal settings. A detailed examination of these unique parameters falls outside the scope of our current work and will be addressed in future research efforts.

### C.1.2 Details about Settings

We performed experiments in the *MiniGrid* environment, focusing on the *MiniGrid-KeyCorridorS3R3-v0* nd *MiniGrid-FourRooms-v0* scenarios. We set up the two environments with a maximum step limit of 300 and used the Adam optimizer with a learning rate of $3 \times 10^{-3}$. The other parameter settings are consistent with those in Table 7.

| Categories | Strategies | Default Settings in LightZero |
|---|---|---|
| MCTS Unique Exploration Strategies | • Adding Dirichlet noise at the root node (also known as the decision node).
• Balancing the weight of prior probability $P$ and MCTS-derived $Q$ estimates at intermediate nodes using the PUCT formula [31]. | • The alpha value in the Dirichlet distribution is 0.3. The noise weight is 0.25.
• The base constant $c_1$=1.25, the initialization constant $c_2$=19652. |
| General Exploration Strategies in RL | • Data collection phase:
  • Decaying temperature coefficient for visit count distribution.
  • Epsilon-greedy exploration strategy.
• Learning phase:
  • Policy entropy regularization.
  • Intrinsic exploartion, e.g., NGU [28].
  • Imitation learning techniques, e.g., Efficient Imitate [72]. | • The temperature transitions from 1 to 0.5 and then to 0.25 in a fixed traininng steps.
• By default, not used.

• Default policy entropy loss weight is 0.

• NGU [28]

• N/A |

Table 2: Overview of *Exploration Strategies* in MCTS algorithms implemented in *LightZero*.

Here, we provide detailed explanations of the various enhanced exploration strategies used in Figure 5 of Section 5.1.

- **Naive**: This is the default exploration strategy, which includes a manual temperature decay mechanism. The initial temperature is 1, and the temperature decays twice during training, to 0.5 and 0.25, respectively. The decay timings are controlled by a parameter. At $50\%$ of this parameter's value multiplied by the training steps, the temperature decays to 0.5. At the $75\%$ point, the temperature decays to 0.25. Afterward, the temperature remains fixed at 0.25.

- **NaiveDoubleSimulation**: Same configuration as Naive except use double number of simulations.

- **FixedTemperature-1**: A fixed temperature parameter was utilized throughout the training period, with the specific value set to x=1.

- **PolicyEntropyRegularization-x**: An additional policy entropy regularization term was added to the original loss of MuZero [14] algorithm, with policy entropy loss weights of 0.05 or 0.005.

- **EpsGreedy**: Inspired by common exploartion settings in value-based RL methods [4], actions were selected uniformly with probability $\epsilon$, and with probability $1 - \epsilon$ with the argmax operation.

- **IntrinsicExploration**: An additional fixed target network and predictor network were introduced. The predictor network was employed to predict the output of the target network, and the normalized MSE loss was used to obtain the intrinsic reward. For specific details, please refer to Section C.1.3.

### C.1.3  Details about Intrinsic Exploration

The core idea of intrinsic rewards is to encourage the agent to visit novel states as much as possible throughout the learning process or from a life-long perspective. In theory, any long-term novelty estimator can be used to generate this incentive. The RND (Random Network Distillation) [27] algorithm demonstrates good performance, is easy to implement, and can be parallelized, so we use it as an example to generate intrinsic rewards, denoted as $r^i$.

**Original MSE loss** Specifically, it utilizes two neural networks:(1) *Random Network*: $g : \mathcal{O} \to \mathbb{R}^k$, a fixed network with randomly initialized parameters that takes the observed observation $x_t$ as input and outputs its encoding $g(x_t)$. (2) *Prediction Network*: $\hat{g} : \mathcal{O} \to \mathbb{R}^k$, a network that takes the observation $x_t$ as input and outputs the predicted value $\hat{g}(x_t; \theta)$ for the observation encoding $g(x_t)$. The networks are trained on the data collected by the agent using stochastic gradient descent, updating the parameters $\theta$ by minimizing the mean squared error $err(x_t) = \|\hat{g}(\mathrm{x}; \theta) - g(\mathrm{x})\|^2$. The fundamental concept of this approach is to leverage the neural networks' ability to model dataset distributions in a manner akin to supervised learning. A larger prediction error for a particular observation suggests that the agent has visited the surrounding observations in the observation space less frequently, indicating a higher degree of novelty for that observation.

**Reward normalization** To mitigate the impact of significant magnitude variations in the mean squared error across training and different environments, the final intrinsic reward is defined using min-max normalization. The combined reward at time $t$ is then defined as: $r_t = r_t^e + \beta r_t^i$, where $r_t^e$ denotes the external reward provided by the environment at time $t$, $r_t^i$ represents the normalized intrinsic reward generated by the exploration module, and $\beta$ is a positive number serving as the weight factor for the intrinsic reward. In this paper, we set $\beta$ to $\frac{1}{300}$ to ensure that the sum of intrinsic rewards in a single episode is less than the maximum original external reward of 1, thus preventing intrinsic rewards from dominating the original objective.

**Key designs** In the specific implementation, a critical aspect is the selection of the input $x_t$. In our preliminary experiments, we used the latent state from the *MuZero* model but observed subpar performance. We hypothesize that this is due to the distribution of latent states changing throughout the training process, causing the target values obtained by the random network to continuously shift and resulting in intrinsic rewards that essentially resemble noise. Ultimately, we chose to use the environment's original observation $o_t$ as the input observed state and achieved satisfactory results.

It is worth mentioning that combining efficient exploration methods, such as intrinsic rewards, with the unique MCTS exploration strategies detailed in Table 2, represents an intriguing research direction. We leave this integration as a potential area for future investigation.

## C.2 Alignment in Environment Model Learning

**Alignment** In this section, we examine case studies centered on the essential objective functions. Drawing inspiration from the paper [25], the importance of alignment in training the representation, policy and model is emphasized. The policy should only access accurate model states, while the representation must encode task-relevant and predictable information. The consistency loss proposed in [17] serves as a candidate technique: at each unroll step, the latent state generated by the dynamic model should be *aligned* (similar/consistent) to the latent state directly obtained from the original observation via the representation network.

**Setting** To investigate the impact of self-supervised consistency loss on different types of observation data, we use the MuZero algorithm as our baseline. The Adam optimizer is employed with a learning rate of $3 \times 10^{-3}$. The notation "MuZero w/ SSL" signifies that the consistency loss weight is 2, while "MuZero" means the consistency loss weight is 0. The other hyperparameters remain the same as those described in Section G.4.

**Special case in board games** As shown in Figure 14, for special image input environments (checkerboard with discrete values) like *TicTacToe*, the cosine similarity relatively low, around 0.662. This highlights the challenge of achieving consistency between the latent state output from the dynamic model and the latent state derived from the observation via the representation network. Since adjacent observations differ only in one position, adding consistency loss with inappropriate optimization settings might hinder the learning progress of other parts of the algorithm. To verify this, we conducted an experiment on *TicTacToe* using the SGD optimizer and compared the performance with and without the consistency loss, as shown in the right panel of Figure 14. We observed that the addition of consistency loss hampers the learning progress in the early stages, which further supports our conjecture. That is to say, our experiments demonstrate that the consistency loss proposed in

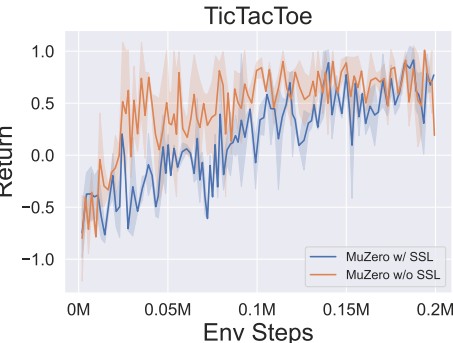

| Env. Name | TicTacToe | LunarLander | Pong |
|-----------|-----------|-------------|------|
| *cos* (0M) | 0.782 | 0.913 | 0.875 |
| *cos* (0.4M) | 0.678 | 0.971 | 0.973 |
| *cos* (0.8M) | 0.662 | 0.974 | 0.987 |

Figure 14: **Left**: The change of cosine similarity (i.e. negative consistency loss, *cos=1.0* indicates that the two vectors have the same direction.) for three input types throughout the MuZero training. In special image input environments such as *TicTacToe*, characterized by checkerboard patterns with discrete values, the cosine similarity between the latent state output from the dynamic model and the latent state derived from the observation via the representation network remains relatively low. This emphasizes the challenge of attaining consistency between these two latent states. **Right**: Effects of self-supervised consistency loss on board games when using SGD [73] optimizer with momentum. The inclusion of consistency loss impedes the learning progress during the initial stages.

[17] is suitable only for environments with specific attributes. For board games, devising a suitable alignment constraint to ensure consistent model learning remains a topic for future research.

# D Explanation for Figure 2

In this section, we will introduce the score rules in Figure 2. We categorize the critical capabilities of general decision solvers into the following dimensions: multi-modal observation space, complex action space, inherent stochasticity, reliance on prior knowledge, simulation cost, hard exploration, and data efficiency. We then compare four algorithms including Model-free RL (e.g. PPO), AlphaZero, MuZero, and LightZero. Please note that in this section, the term *LightZero* refers to the algorithm that embodies the optimal combination of techniques and hyperparameter settings within our framework. We will elaborate on the specific interpretation of each algorithm depicted in Figure 2.

**Model-free RL (PPO)** We use the standard on-policy PPO implementation and DI-engine [74] benchmark results. This implementation includes optimized hyper-parameters and incorporates various techniques to enhance performance across different environments. The list of typical hyperparameters for PPO is presented in Table 3.

| Hyperparameter | Value |
|----------------|-------|
| Epoch per collect | 10 |
| Num of samples per collect | 3200 |
| Batch size | 320 |
| Discount factor | 0.99 |
| GAE lambda | 0.95 |
| Recompute advantage | True |
| Entropy weight | 0.001 |
| Dual clip | True |
| Gradient norm | 0.5 |
| PopArt | True |

Table 3: Key hyperparameters of model-free PPO methods.

**AlphaZero** We use the default implementation of AlphaZero provided by LightZero, following the settings from the original AlphaZero paper [29].

**MuZero** We employ the default implementation of MuZero provided by LightZero, following the settings described in the original MuZero paper [14] and the public codebase MuZero-General [51].

**LightZero** We have incorporated the relevant MCTS/MuZero algorithm techniques, each tailored for diverse environments with unique challenges. This integration has given rise to a specialized algorithm version, dubbed *LightZero*, within our framework.

The score assigned to each algorithm in the distinct dimensions indicates their performance and applicability. A score of 1 suggests poor performance and limited applicability, while a higher score implies a broader application scope and better performance. That is to say, a score of 2 means that this algorithm can deal with some parts of envionments or problems in this dimension. A score of 3 means that it can tackle more than a half of issues while 4 indicates it has already been capable of many challenges but there is still some improved space. The highest score 5 is assigned to those methods that could be state-of-the-art results in this dimension or don't need to care about this problem. We conduct a qualitative analysis to compare the following algorithm versions.

**Multi-modal Observation Space** AlphaZero is customized to model the 2D image-like observation in board games, receiving a score of 2. MuZero extends this capability and is able to handle more general 2D image observations including video games. Model-free RL methods and LightZero utilize extra self-supervised learning representation learning techniques, enabling them to work effectively with high-dimensional and complex states. However, challenges still exist in learning abstract decision behaviors within multi-modal representations.

**Complex Action Space** In terms of action spaces, AlphaZero is limited to discrete actions in board games. MuZero extends it to various discrete action spaces. Model-free RL methods are also able to handle both discrete and continuous actions. However, in more complex hybrid action space, these methods often require special mechanisms, such as the autogressive action prediction used in AlphaStar [3]. Sampled MuZero represents a significant endeavor to implement MCTS in continuous action spaces, employing finite action samples as the primary basis for action selection. However, when the number of samples is insufficient, performance may suffer due to inadequate approximation. Despite this, LightZero, which incorporates this sampling mechanism, achieves a score of 4, effectively showcasing its ability to address these challenges.

**Inherent Stochasticity** Ths stochasicity of environment dynamics poses a significant challenge for RL algorithms. Conventional MCTS algorithms face planning issues due to inconsistent state transitions. Value-based RL methods, such as DQN, often face difficulty adapting to variable reward signals for identical actions within a particular state, leading to suboptimal performance. Actor-Critic algorithms like PPO somewhat mitigate this issue. LightZero adopts insights from [16] and [75], employing an enhanced model that simultaneously learns the distribution of latent chances for various scenarios. These learned probabilities are subsequently incorporated into the tree search nodes. While these adjustments have bolstered MCTS's ability to handle inherent dynamics randomness and the partially observable state, the challenge remains substantial when addressing real-world decision-making scenarios.

**Reliance on Prior Knowledge** One key limitation of AlphaZero is its reliance on prior knowledge of environments, such as the perfect simulator or game rules and records. MuZero addresses this limitation by designing a dynamics model that learns the reward and transition function, significantly relaxing this requirement. For model-free RL methods and LightZero, they are more flexible but still require some prior-oriented operations like observation pre-processing and reward shaping, to transform the original problem into a more standard MDP. Therefore, a score of 4 is suitable for them.

**Simulation Cost** Although MCTS-style methods show great data effciency, it is important to acknowledge that the wall-time of related training programs can be astonishing due to the simulation cost. These methods necessitate the establishment of comprehensive search trees to assure adequate visit counts and precise value estimation LightZero, by integrating core ideas from the Gumbel MuZero paper and utilizing other optimization techniques, effectively minimizes this overhead. However, due to its inherent model-based nature, it still demands a longer runtime compared to model-free RL methods.

**Hard Exploration** Tree-search-based planning methods enhance exploration capabilities by reducing the search space for the optimal policy. Conversely, traditional model-free RL methods often struggle in sparse reward environments without specific techniques. AlphaZero gains a slight advantage in this dimension by utilizing a perfect simulator rather than models that are incrementally trained over time. LightZero incorporates additional exploration strategies at minimal cost and successfully tackles the hard exploration task efficiently (such as *MiniGrid*).

**Data Efficiency** MCTS-style methods excel in data efficiency compared to model-free RL methods, offering impressive performance on academic benchmarks like Atari-100K and DMControl-500K. LightZero implements specialized data sampling with staleness and reuse limitation and management mechanisms (e.g. throughput limiter) to further improve data efficiency.

# E    Efficiency Analysis in LightZero

This section first lists the computational hardware configurations of all the experiments. For nearly all basic experiments and ablations, we aim to validate the runtime efficiency on a small resource budget, thus we deploy each experiment instance on the Kubernetes cluster with computational resource of 1 A100 40G GPU, 24 CPU cores and 100GB RAM. Given these settings, LightZero can train Atari agents for 100K steps in 4 hours, and it can conduct 100K steps of self-play on Gomoku in 5 hours. Furthermore, we profile the entire training pipeline of these tiny instances. Based on these analysis, we can identify several bottlenecks in the runtime of the MCTS/MuZero training pipeline and found some practical, albeit imperfect, solutions to these bottlenecks.

**Environment Latency** RL environments usually vary in the execution time of *reset* and *step* methods. For AlphaZero-style methods, environment's *step* method are called many times (e.g. more than 50 in one action selection) and lead to the huge time cost. Particularly, AlphaZero deployed on board games may spend more than a half of the whole training time in environments. In practice, we use the *vmap* and *jit* mechanisms provided by JAX [76] and the LRU (Least Recently Used) cache trick to accelerate these methods. Thanks to many for-loop and duplicate operations in classic board game implementations, we can significantly reduce the associated overhead and efficiently implement self-play training.

**Tree Search** Although the tree search process doesn't contain many complex mathematical operations, it is still a non-neglected component in the efficiency optimization. Moreover, the naive vectorized environment scheme shows no obvious gain in tree search methods because each envionments needs a unique search tree, which is not siutable to use the batch processing to speed up. On the other hand, due to the large number of call times, some basic python primitives like *getattr* and *setattr* have become a drawback of this module, thus LightZero implements a variety of tree search operations in MCTS/MuZero algorithm variants with cpp and Cython extension for Python, and it offers obvious improvements in various fine-grained code blocks.

**Model Inference** As MuZero-style methods utilize a learned model in place of the perfect simulator, they spend more time on the inference procedure of neural network models. To tackle the efficiency issues associated with more complex network architectures and larger network parameters, LightZero leverages various tools from the open-source community, including some features of PyTorch 2.0 [77], some tricks about mixed precision training and large batch training in RL.

**Parallel MCTS** Utilizing a large CPU cluster and enabling multiple parallel data collectors has proven successful in many decision-making tasks [9, 8]. When it comes to MCTS-style methods, due to the difficulty mentioned in tree search part, previous methods [14, 34] mainly use the collector-parallel scheme, each collector is composed of an environment, a search tree and a RL agent. Naive batch processing inside the collector can lead to significant waiting time between the three components: env step time, model inference time and tree search time (shown in Figure 15 (d)), which severely underutilized computational resources. As suggested in Figure 15(e), LightZero launches multiple (k) environments and search trees in a collector, but divides these modules into several groups, the number of group should be determined by three concrete time counts for a specific environment.

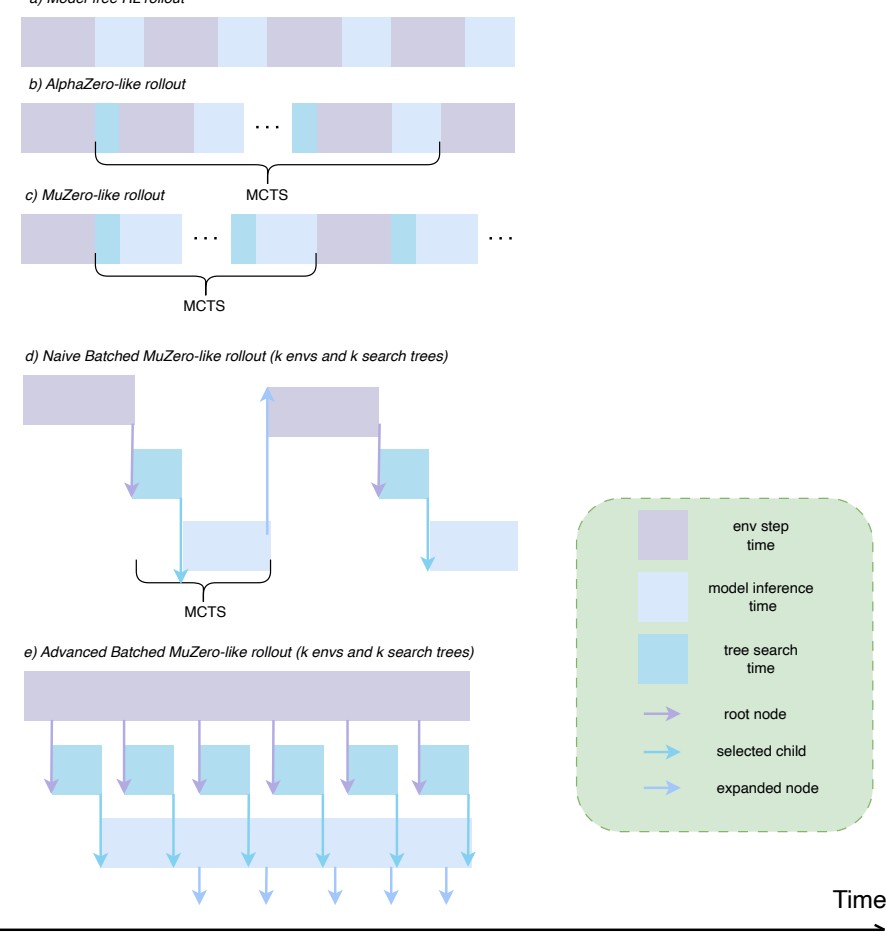

Figure 15: Comparison of various data collection pipelines. a), b), c) illustrate simple serial pipelines for different types of algorithms. d) and e) depict two distinct parallel MCTS schemes, the former will cause the obvious waiting time while the latter can utilize all types of computation resources as much as possible, which makes three different parts overlapped with each other.

Specifically, data collector alternate between different groups to execute corresponding steps. For example, even when the first group of environments has completed its steps, the corresponding CPUs are not idle as they can execute environment simulation steps for the second group. Correspondingly, RL agents can use the similar strategies to improve the utilization on GPU. Although the ideal case illustrated in Figure 15(e) is challenging to achieve perfectly, this grouping and buffering mechanism can significantly enhance the parallel capabilities of MCTS in practical data collection.

**Multi-GPU Training** Leveraging the Distributed Data Parallel module from PyTorch, we've implemented multi-GPU training capabilities into the LightZero codebase. This functionality has been rigorously tested with the *EfficientZero* model within the *Atari PongNoFrameskip-v4* environment, using 1, 2, and 4 GPUs. The results, depicted in Table 4, demonstrate that by training with 4 GPUs, we observe an approximate five-fold increase in speed, while maintaining similar levels of performance. Consequently, with access to additional GPU computing resources, we anticipate an even greater acceleration of the training process.

**Reanalyze** From the viewpoint of system design, data reanalyze is the most special component in MuZero-style methods. SpeedyZero [34] explores the necessity and best practice of separate reanalysis nodes in distributed training. They use different model replicas to compute priority for sampling and new targets for training. LightZero optimizes the cost of reanalyze modules with a

| Number of GPUs | Approximate Time to 1M Env Steps (mins) |
|:---:|:---:|
| 1 | 844 |
| 2 | 363 |
| 4 | 152 |

Table 4: Approximate training times for *EfficientZero* on the *PongNoFrameskip-v4* environment with different numbers of GPUs. The results show that training with 4 GPUs can improve the speed by about 5 times with similar performance, which is in line with our expectations.

simple yet effective selection mechanism: only the data with high training potential (e.g., appropriate state-action novelty and high temporal difference error) and sufficient training stability require a high-frequency reanalysis ratio; other data can undergo fewer reanalysis operations to save time.

**Communication** Transporting model and data is two main communications in the distributed RL training. To ensure the stable convergence of agents, the entire training pipeline cannot collect excessive data or training the network two many times. The former shows few performance gain or even lead to some harmful training batch, while the latter can result in severe over-fitting. According to the actual speed of sending models (from agent learner to data collector and data arranger) and sending collected data, LightZero designs a throughput limiter to control the ratio between generating new data and sample training batch. By using a fixed batch size, we can adjust the ratio within a reasonable range (such as 0.8x-1.2x of original settings). Besides, there is a unnecessary dataflow of the RL training pipeline, which is described in Figure 16. In classic RL frameworks, the latest generated data in data collector is located in the inference GPU, however, it has to go through many steps to become serialized bytes data and then it is sent to agent learner by normal communication techniques like socket. To address this problem, we utilize the RDMA (Remote Direct Memory Access) technique to send the training data from inference GPU to training GPU directly. With the aid of this improvement, LightZero can expedite data collection in the P2P (Peer-to-Peer) communication mechanism, bypassing a series of previously complicated operations.

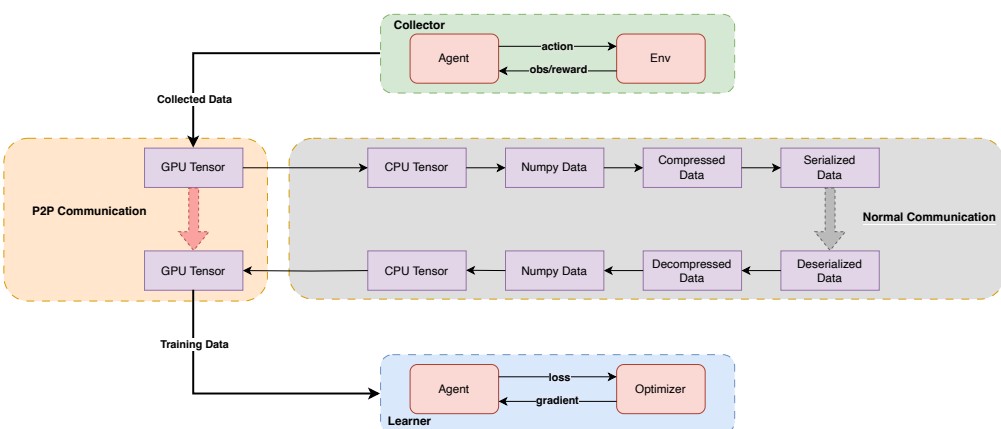

Figure 16: Comparison of various algorithms during the data collection phase. Utilizing Peer-to-Peer (P2P) communication with the Remote Direct Memory Access (RDMA) technique significantly reduces the time required in the data collection process, bypassing the complex operations typically employed in conventional communication methods.

# F   Detailed sub-modules description of MuZero

To elucidate the application of specific algorithms into the various sub-modules that constitute LightZero, we delve into the implementation of the MuZero algorithm. This discussion is structured around the sub-modules as outlined in Figure 3:

**Data Collector** MuZero is representative of online Reinforcement Learning (RL) algorithms, demanding substantial interactions between the agent and the environment to amass training data. Within this sub-module, we leverage a vectorized environment manager and a deeply optimized batch-parallel search tree, facilitated by Cython/C++ extensions. These elements work collaboratively to ensure a high-throughput data collection process.

**Data Arranger** The function of the Data Arranger sub-module is to maximize the utility of different stale data. In the context of MuZero, the first step entails storing complete trajectories or episodes in a prioritized replay buffer to maintain their temporal sequence. Subsequently, we institute a priority recomputer mechanism to periodically determine the sampling priority of the stored data. The calculated priority is proportional to the likelihood of the data being sampled for training. Furthermore, to rectify any off-policy bias and enhance the stability of training, a data reanalyzer updates the target value stored in data, utilizing the latest network parameters. Recognizing the intricate "producer-consumer" relationship between the Data Collector and Agent Learner, MuZero incorporates a throughput limiter to monitor the number of push/pop data operations and regulate the allocation of computational resources.

**Agent Learner** The Agent Learner sub-module amalgamates an array of deep learning and reinforcement learning techniques to train a set of neural networks as defined in MuZero. Components of this sub-module include the distributional RL module, which models the inherent randomness of environmental reward, and the data parallel and mixed precision training utilities that expedite per-iteration time. These features can be conveniently enabled or disabled via the corresponding configuration fields.

**Agent Evaluator** Throughout the training process, MuZero necessitates the evaluation of the performance of the newly trained network. The Agent Evaluator sub-module encompasses several evaluation strategies such as low-temperature sampling and beam search to enhance results. This sub-module also employs a broad spectrum of metrics and visualizations for agents.

**Context Exchanger** To achieve asynchronous execution across four sub-modules and to efficiently scale the entire training pipeline, we employ a Context Exchanger. This component incorporates novel communication elements to facilitate the efficient transfer of necessary context information.

By dissecting the implementation of the MuZero algorithm into these sub-modules, we demonstrate the flexibility and modularity of LightZero, and how it can be effectively employed in the design and execution of complex RL algorithms.

## G   Details about Algorithm Implementation

In this section, we first provide an overview of the subsequent algorithm extensions of MuZero, followed by a comprehensive algorithm overview diagram illustrating the implementation of various integrated algorithms in LightZero (e.g., Figure 19). Next, we introduce the model's network architecture, including the representation, dynamics, and prediction network. Finally, we present the hyperparameter settings for each algorithm and environment used in the baseline results.

### G.1   MuZero's Extensions

In recent years, MuZero has been extended through various algorithmic innovations to enhance its efficiency and stability across different scenarios. These extensions mainly include:

**Sampled MuZero** [45] This approach introduces a general framework called the sample-based policy iteration, which can be theoretically applied to any types of action spaces. The core idea is to compute improved policies within a subset of the original action space. As the number of sampled actions gradually increases towards the size of the entire action space, the sampled improved policy converges probabilistically to the improved policy over the entire action space.

**Gumbel MuZero** [18] is an extension designed to enhance performance in environments with low simulation costs by leveraging the Gumbel-Top-k trick [78] to select actions that guarantee policy

improvement. This approach introduces a new improved policy, $\pi'$, which seamlessly integrates the original visit counts distribution with MCTS searched values. By combining these two sources of information, Gumbel MuZero efficiently exploits the knowledge acquired during the search process, leading to more informed decision-making and improved performance in a variety of environments.

**Stochastic MuZero** [16] This extension enhances MuZero by enabling it to learn and plan with stochastic models. Specifically, it incorporates a stochastic model that includes afterstates and conducts stochastic tree searches using this model. Stochastic MuZero achieves competitive or superior performance compared to state-of-the-art methods in a variety of canonical board games. such as 2048 and Backgammon, while preserving the superhuman performance demonstrated by the standard MuZero in Go.

## G.2 LightZero Algorithms Overview

In this section, we present a detailed summary of various MCTS/MuZero variants, encapsulated in algorithm overview diagrams. Specifically, Figure 17 provides an insight into the Monte Carlo Tree Search (MCTS), while Figure 18 illuminates the mechanics of AlphaZero, a fusion of MCTS and deep neural networks. Moreover, Figure 19 introduces MuZero, an advancement upon AlphaZero with a learned model representation. Figure 21 subsequently presents the Sampled MuZero, and finally, Figure 22 delineates the Gumbel MuZero variant. These overview diagrams serve to highlight the evolution and interconnections among MCTS, AlphaZero, MuZero, and their related algorithms.

## G.3 Model Architecture

We provide the representation network structure of the MuZero series algorithms implemented in LightZero in Figure 23. The dynamics network and prediction network structures (including both value and policy networks) are presented in Figure 24. Note that the model structures we provide are based on Atari environments with image-type observations. For environments with vector-type observations, such as continuous control tasks, the overall model structure is similar, only replacing the convolutional layers in the original model with corresponding fully connected layers. For specific details, please refer to the model directory through OpenDILab at https://github.com/opendilab/LightZero.

## G.4 Hyperparameters

We provide a detailed summary of the key hyperparameters for the *MuZero w/ SSL* algorithm, as implemented in LightZero for the *Atari* environment, in Table 7. The hyperparameters for other environments and algorithms are generally similar. Unless explicitly stated otherwise, all other parameters are in accordance with those in Table 7.

Specifically, we highlight the primary differing hyperparameters for the *Sampled EfficientZero* algorithm in the *Atari* environment in Table 8, as well as for continuous control environments such as MuJoCo in Table 9. In addition, we explain the main distinct hyperparameters for the *Gumbel MuZero* algorithm in the Atari environment in Table 10, alongside the *Stochastic EfficientZero* algorithm in both the Atari environment, as shown in Table 11, and the *2048* environment, as presented in Table 12. Lastly, the pivotal hyperparameters for the *AlphaZero/MuZero* algorithm in the *Gomoku* environment are set out in Tables 13 and 14.

## H Comparison with Other MCTS Frameworks

*LightZero* is a comprehensive algorithm benchmark developed using PyTorch, integrating nine distinct algorithms such as AlphaZero and MuZero. It supports over twenty different environments, including Atari and Go. On the other hand, the *MCTX* library[2], primarily implemented using JAX, includes basic implementations of algorithms such as AlphaZero, MuZero, and Gumbel MuZero. However, the training process across various environments is not yet fully established.

---

[2] https://github.com/google-deepmind/mctx

To offer an intuitive comparison of the differences in integrated algorithms and supported environments between these two repositories, we present the algorithms and environments supported by LightZero and MCTX in the following Table 5 and Table 6, respectively.

Please note: "✓" denotes that the corresponding item is fully implemented and thoroughly tested. "⊠" signifies that the item is currently in development. "—" indicates that the algorithm does not support the respective environment.

Regarding the comparison table provided, we should note the following two points: Firstly, the list of algorithms and environments supported by MCTX not only encompasses MCTX itself, but also includes all derivative repositories based on MCTX. Secondly, all environments associated with Gumbel MuZero and Stochastic MuZero within the MCTX library are currently in a locked "⊠" state. This is because, to our knowledge, neither MCTX nor its derivative repositories have fully implemented these algorithms and environments. Although the foundational modules for the Gumbel MuZero and Stochastic MuZero algorithms exist in the original MCTX repository, the development of a complete training pipeline for these algorithms is still in progress.

| Algo Env | Classic Control | Box2D | Atari | MuJoCo | Go Bigger | Mini Grid | Maze | Connect Four | Gomoku | 2048 | Go |
|---|---|---|---|---|---|---|---|---|---|---|---|
| *AlphaZero* | — | — | — | — | — | ⊠ | ⊠ | ✓ | ✓ | ⊠ | ✓ |
| *Sampled AlphaZero* | — | — | — | — | — | ⊠ | ⊠ | ⊠ | ✓ | ⊠ | ✓ |
| *Gumbel AlphaZero* | — | — | — | — | — | ⊠ | ⊠ | ⊠ | ✓ | ⊠ | ✓ |
| *MuZero* | ✓ | ✓ | ✓ | — | ✓ | ✓ | ✓ | ✓ | ✓ | ✓ | ✓ |
| *MuZero w/ SSL* | ✓ | ✓ | ✓ | — | ✓ | ✓ | ⊠ | ⊠ | ✓ | ✓ | ✓ |
| *EfficientZero* | ✓ | ✓ | ✓ | — | ✓ | ✓ | ⊠ | ⊠ | ✓ | ✓ | ✓ |
| *Gumbel MuZero* | ✓ | ✓ | ✓ | — | ✓ | ✓ | ⊠ | ⊠ | ✓ | ✓ | ✓ |
| *Sampled MuZero* | ✓ | ✓ | ✓ | ✓ | ✓ | ✓ | ⊠ | ⊠ | ✓ | ✓ | ✓ |
| *Stochastic MuZero* | ✓ | ✓ | ✓ | — | ✓ | ✓ | ⊠ | ⊠ | ✓ | ✓ | ✓ |

Table 5: Algorithms and Environments supported by *LightZero* as of the writing of this paper.

| Algo Env | ClassicControl | Box2D | Atari | Maze | ConnectFour | Gomoku | Go |
|---|---|---|---|---|---|---|---|
| *AlphaZero* | — | — | — | ✓ | ✓ | ✓ | ✓ |
| *MuZero* | ✓ | ✓ | ⊠ | ⊠ | ⊠ | ⊠ | ⊠ |
| *Gumbel MuZero* | ⊠ | ⊠ | ⊠ | ⊠ | ⊠ | ⊠ | ⊠ |
| *Stochastic MuZer* | ⊠ | ⊠ | ⊠ | ⊠ | ⊠ | ⊠ | ⊠ |

Table 6: Algorithms and Environments supported by *MCTX* as of the writing of this paper.

# I  Limitations and Future Work

Despite its considerable achievements across various benchmark environments, LightZero does bear certain limitations.

① Firstly, while efforts have been made to amplify the algorithm's universality through decoupling and modularization, certain specific problems may still necessitate tailored adjustments and optimizations. For instance, all current LightZero algorithms do not directly support hybrid action space environments [79]. However, through the use of action representation [80] and other techniques, they could potentially be adapted to a wider action space.

② Secondly, as discussed in Section 3, due to the intrinsic limitations of the MCTS algorithm, our method may encounter challenges in specific, complex real-world environments. Particularly when handling environments with high-dimensional action spaces or strong randomness, methods like *Sampled MuZero* and *Stochastic MuZero* could benefit from further optimization and improvement.

③ Finally, the high prerequisites of LightZero can pose challenges, especially for those new to decision intelligence. This is largely due to the inherent complexity of the MCTS+RL algorithms. While these algorithms are comprehensive, they require a profound understanding and significant time investment to deploy effectively. This complexity, alongside the computational demands of running these advanced algorithms, can limit their widespread application. Future enhancements should concentrate on improving the accessibility of the LightZero framework's interfaces, enriching the documentation, and cultivating a vibrant user community that encourages knowledge collaboration.

Despite the aforementioned challenges, we remain optimistic about the potential of the LightZero framework. For future, we have identified several promising areas for further exploration:

- **Broadening Applications**: We envision more researchers and developers applying LightZero across a wider spectrum of practical fields. These include but are not limited to natural language processing, autonomous driving, and the control and optimization of complex systems.

- **Algorithmic Optimization**: While we have enhanced our submodules by introducing more suitable exploration mechanisms and optimizing model loss functions, there remains significant room for improvement. We invite the community to contribute new exploration and optimization strategies to further boost the performance of MCTS-style algorithms.

- **Integration with Cutting-Edge Technologies**: We stress the importance of a seamless integration between LightZero and future research directions. Two key areas for future exploration stand out:

    1. *Integration with Large Language Models (LLM)* [81] [82]: LLM can serve as a high-level task planner, breaking down complex decision-making problems into a series of low-level instructions. LightZero can set these instructions as conditional inputs of the representation network and serve as an efficient instruction executor. Furthermore, considering the powerful reanalysis mechanism of MuZero Unplugged, it can be a viable choice for fine-tuning LLMs.

    2. *Clever Utilization of Model-Based RL Techniques*: LightZero has already integrated many model-based RL techniques, such as the Recurrent State-Space Models (RSSM) [83] used in DreamerV2 [84], latent state consistency loss used in EfficientZero, and the afterstate dynamics functions proposed in Stochastic MuZero [16]. We aim to incorporate additional techniques within the model representation, loss function, and MCTS search process to enhance the universality and robustness of the LightZero framework.

**Monte Carlo Tree Search**

| A. Selection | B. Expansion | C. Evaluation | D. Backpropagation |
|---|---|---|---|
| In each node, select action $a_k$ according to the **UCB** score: $a_k = \arg\max_a \left[ Q(o,a) + c \cdot \frac{\sqrt{\sum_b N(o,b)}}{1+N(o,a)} \right]$ The selection function is applied recursively until a **leaf** node is reached. | The **leaf node** $o_l$ is added to the search tree. Each edge $(o_l, a)$ from the newly expanded node is initialized to: $\{N(o_l, a) = 0, Q(o_l, a) = 0\}$ | The leaf node is evaluated by running a rollout to the **end** of the game with the fast **rollout policy** (potentially random), then computing the winner with function $r$. | Action values $Q$ are updated to track the mean value of all evaluations $\{r\}$ in the subtree below that action. $N$ are updated to track how many times the action has been on the whole MCTS simulations. |

NOTE: The MCTS is divided into **four** stages, and **repeated for a number of simulations**. Once the search is complete, we select the action with the **highest** number of visit.

Figure 17: MCTS algorithm overview.

**AlphaZero**: A general reinforcement learning algorithm that masters chess, shogi, and Go through self-play

**A. MCTS in AlphaZero**

| a. Selection | b. Expansion | c. Evaluation | d. Backpropagation |
|---|---|---|---|
| In each node, select action $a_k$ according to the **UCB** score: $a_k = \arg\max_a \left[ Q(o,a) + cp(o,a) \cdot \frac{\sqrt{\sum_b N(o,b)}}{1+N(o,a)} \right]$ The selection function is applied recursively until a **leaf** node is reached. | The **leaf node** $o_l$ is added to the search tree. Each edge $(o_l, a)$ from the newly expanded node is initialized to: $\{N(o_l, a) = 0, Q(o_l, a) = 0, P(o_l, a = P_l)\}$ | The leaf node is evaluated by the neural network: $p_l, (v_l) = f_\theta(o_l)$ ; the vector of policy priors $P_l$ are stored in the outgoing edges from $o_l$. | Action values $Q$ are updated to track the mean value of all evaluations $\{v\}$ in the subtree below that action. $N$ are updated to track how many times the action has been on the whole MCTS simulations. |

**B. Acting (Self-play)**

**C. Training**

$$l_t(\theta) = l^v(z_t, v_t) + l^p(\pi_t, \mathbf{p}_t) + c\|\theta\|^2$$

where, $z_t$ is the game reward from the **perspective of the current player** and $\pi_{t+k}$ is the MCTS searched policy at timestep $t$.
$v_t$ and $\mathbf{p}_t$ is the predicted value and policy from the neural network $f_\theta$.
$l^v$ is MSE loss, $l^p$ is cross-entropy loss.

Figure 18: AlphaZero algorithm overview.

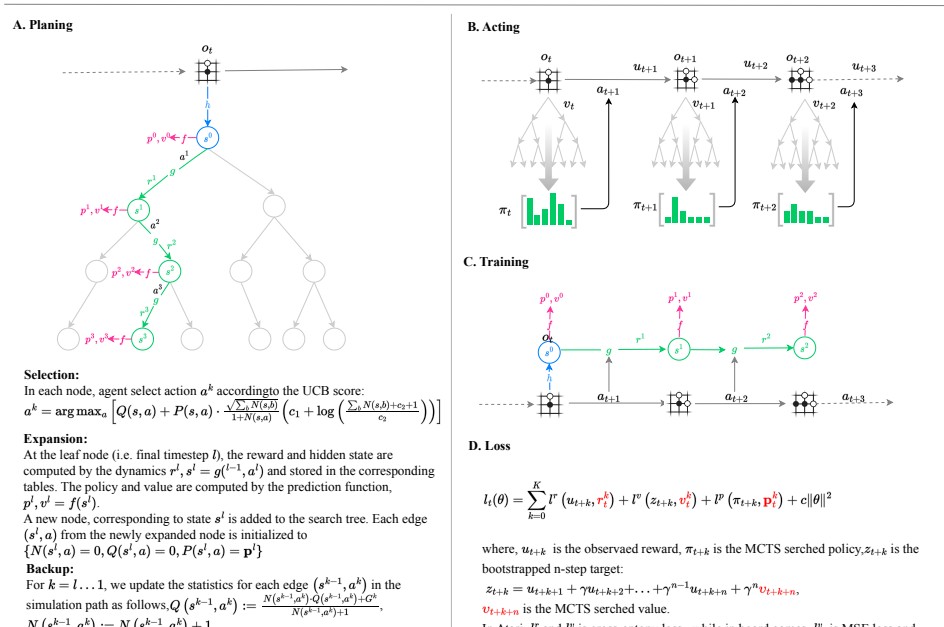

**A. Planing**

**Selection:**
In each node, agent select action $a^k$ accordingto the UCB score:
$$a^k = \arg\max_a \left[ Q(s,a) + P(s,a) \cdot \frac{\sqrt{\sum_b N(s,b)}}{1+N(s,a)} \left( c_1 + \log\left( \frac{\sum_b N(s,b)+c_2+1}{c_2} \right) \right) \right]$$

**Expansion:**
At the leaf node (i.e. final timestep $l$), the reward and hidden state are computed by the dynamics $r^l, s^l = g(^{l-1}, a^l)$ and stored in the corresponding tables. The policy and value are computed by the prediction function, $p^l, v^l = f(s^l)$.
A new node, corresponding to state $s^l$ is added to the search tree. Each edge $(s^l, a)$ from the newly expanded node is initialized to $\{N(s^l, a) = 0, Q(s^l, a) = 0, P(s^l, a) = \mathbf{p}^l\}$

**Backup:**
For $k = l \ldots 1$, we update the statistics for each edge $(s^{k-1}, a^k)$ in the simulation path as follows, $Q(s^{k-1}, a^k) := \frac{N(s^{k-1},a^k)\cdot Q(s^{k-1},a^k)+G^k}{N(s^{k-1},a^k)+1}$,
$N(s^{k-1}, a^k) := N(s^{k-1}, a^k) + 1$,
where, in hypothetical step $k$, we utilize the $l-k$ bootstapped estimate Q value:
$G^k = \sum_{\tau=0}^{l-1-k} \gamma^\tau r_{k+1+\tau} + \gamma^{l-k} v^l$

**B. Acting**

**C. Training**

**D. Loss**

$$l_t(\theta) = \sum_{k=0}^{K} l^r\left(u_{t+k}, r_t^k\right) + l^v\left(z_{t+k}, v_t^k\right) + l^p\left(\pi_{t+k}, \mathbf{p}_t^k\right) + c\|\theta\|^2$$

where, $u_{t+k}$ is the observaed reward, $\pi_{t+k}$ is the MCTS serched policy, $z_{t+k}$ is the bootstrapped n-step target:

$$z_{t+k} = u_{t+k+1} + \gamma u_{t+k+2} + \ldots + \gamma^{n-1} u_{t+k+n} + \gamma^n v_{t+k+n},$$

$v_{t+k+n}$ is the MCTS serched value.
In Atari, $l^r$ and $l^v$ is cross-entropy loss, while in board games, $l^v$ is MSE loss and there is no $l^r$ due to no intermediate reward. $l^p$ is cross-entropy loss for both.

NOTE： $g$ is the dynamic network (MLP, without RNN), value rescale, categorical distribution for reward and value in Atari, Reanalyze

Figure 19: MuZero algorithm overview.

**EfficientZero**: Self-supervised coonsistency loss, Value prefix, Off-policy correction

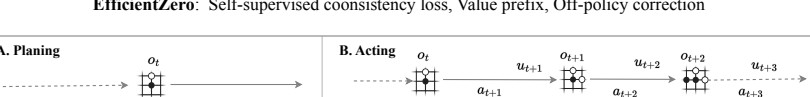

**A. Planing**

**B. Acting**

**C. Training**

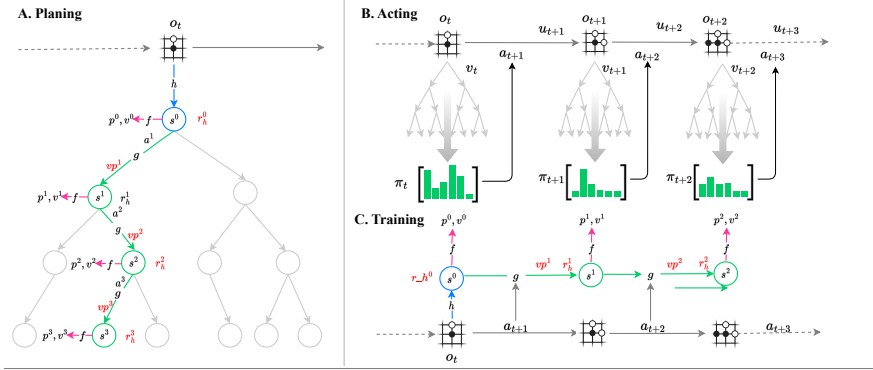

**D. Network and Loss**

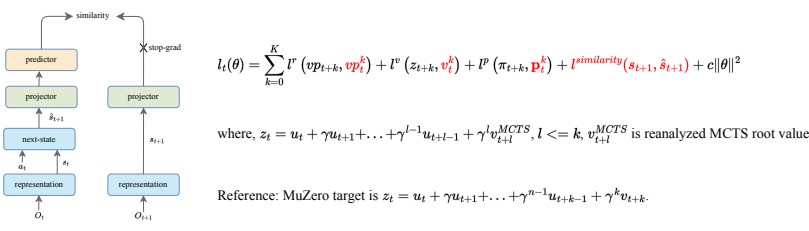

The self-supervised consistency loss.

$$l_t(\theta) = \sum_{k=0}^{K} l^r\left(vp_{t+k}, vp_t^k\right) + l^v\left(z_{t+k}, v_t^k\right) + l^p\left(\pi_{t+k}, \mathbf{p}_t^k\right) + l^{similarity}(s_{t+1}, \hat{s}_{t+1}) + c\|\theta\|^2$$

where, $z_t = u_t + \gamma u_{t+1} + \ldots + \gamma^{l-1} u_{t+l-1} + \gamma^l v_{t+l}^{MCTS}$, $l <= k$, $v_{t+l}^{MCTS}$ is reanalyzed MCTS root value.

Reference: MuZero target is $z_t = u_t + \gamma u_{t+1} + \ldots + \gamma^{n-1} u_{t+k-1} + \gamma^k v_{t+k}$.

NOTE： $g^{rnn}$ (abbreviated as $g$ in the figure) is the *recurrent* dynamics network, *vp* is value prefix, $r_h$ is reward hidden state.

Figure 20: EfficientZero algorithm overview.

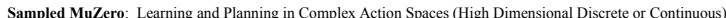

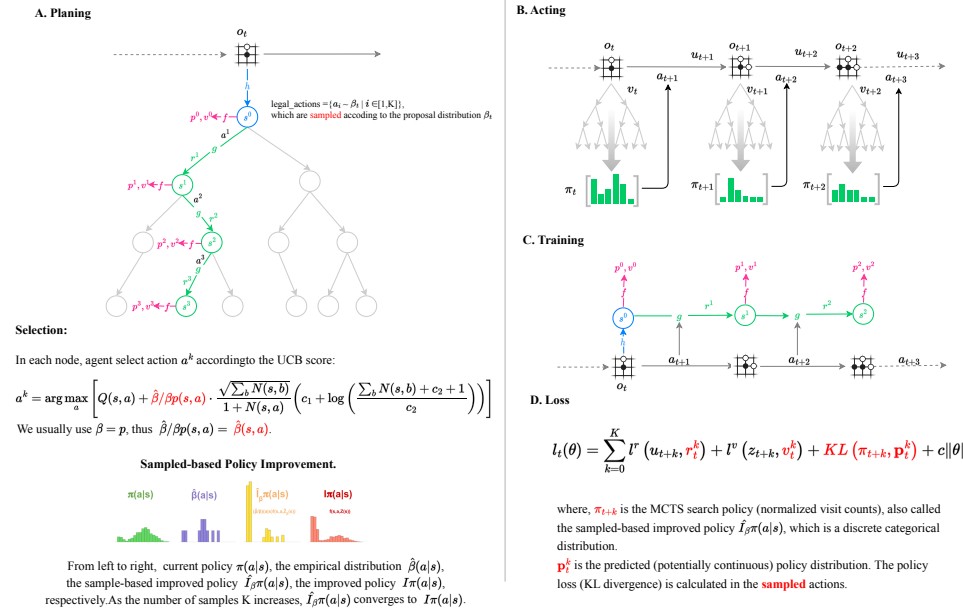

Figure 21: Sampled MuZero algorithm overview.

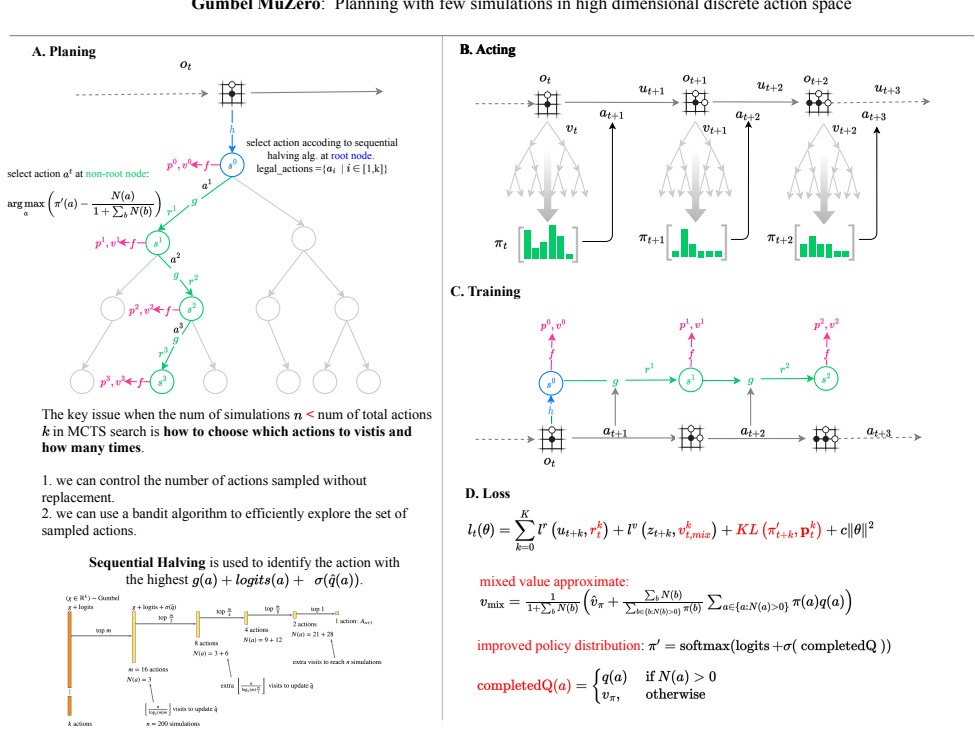

Figure 22: Gumbel MuZero algorithm overview.

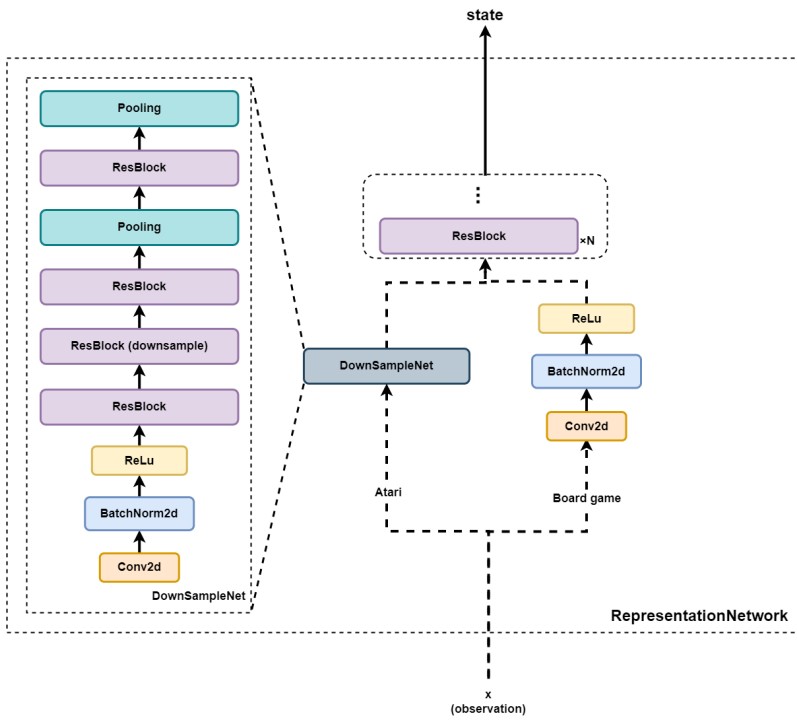

Figure 23: The network architecture of the *representation network h* for image input domain in LightZero. This network represents the root observation as a latent state.

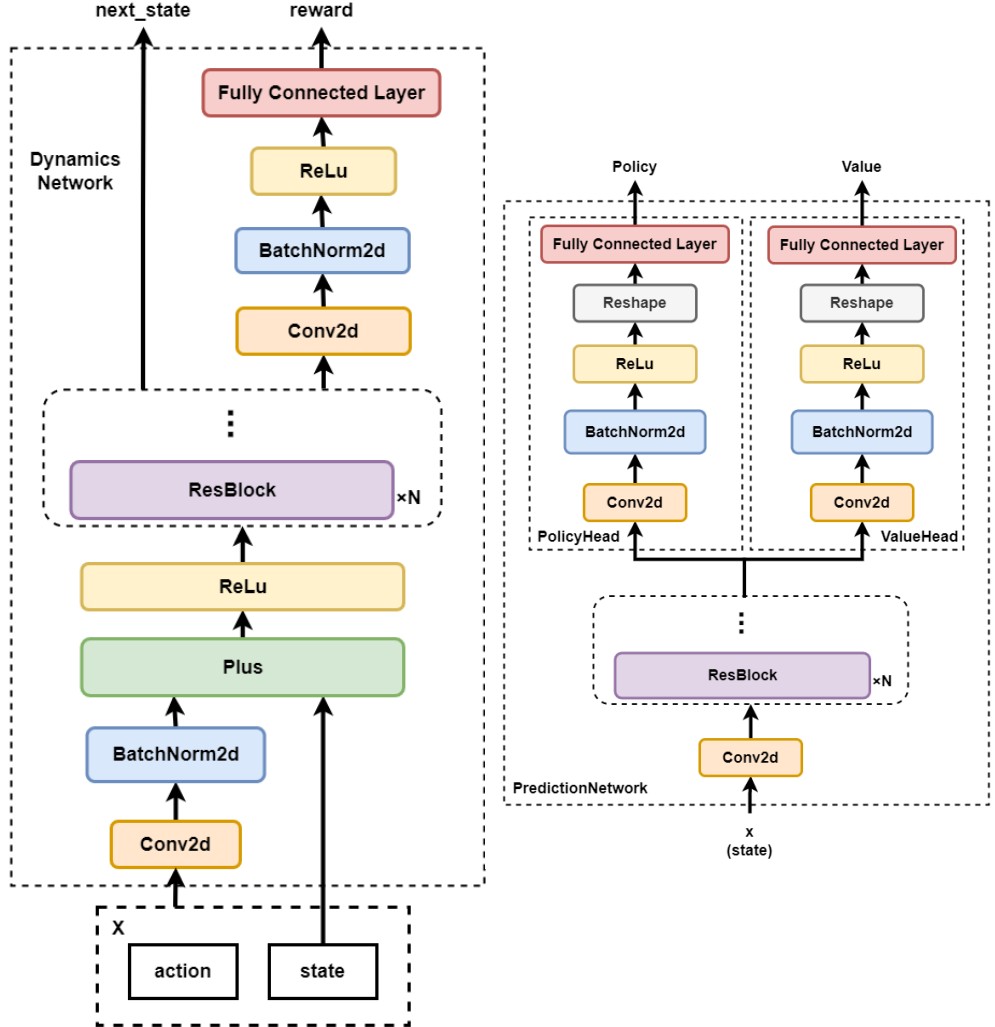

Figure 24: **Left**: The network architecture of the *dynamics network g* for image input domains in LightZero. Given a latent state and a selected action, it outputs the transitioned next latent state and the corresponding predicted reward. **Right**: The network architecture of the *prediction network f* (encompassing both value and policy networks) for image input domains. Given a latent state, this network predicts the action probability and value.

| Hyperparameter | Value |
| --- | --- |
| Num of frames stacked | 4 |
| Num of frames skip | 4 |
| Reward clipping | True |
| Optimizer type | Adam |
| Learning rate | $3 \times 10^{-3}$ |
| Discount factor | 0.997 |
| Weight of policy loss | 1 |
| Weight of value loss | 0.25 |
| Weight of reward loss | 1 |
| Weight of policy entropy loss | 0 |
| Weight of SSL (self-supervised learning) loss | 2 |
| Batch size | 256 |
| Model update ratio | 0.25 |
| Frequency of target network update | 100 |
| Weight decay | $10^{-4}$ |
| Max gradient norm | 10 |
| Length of game segment | 400 |
| Replay buffer size (in transitions) | 1e6 |
| TD steps | 5 |
| Number of unroll steps | 5 |
| Use augmentation | True |
| Discrete action encoding type | One Hot |
| Normalization type | Batch Normalization |
| Priority exponent coefficient | 0.6 |
| Priority correction coefficient | 0.4 |
| Dirichlet noise alpha | 0.3 |
| Dirichlet noise weight | 0.25 |
| Number of simulations in MCTS (sim) | 50 |
| Reanalyze ratio | 0 |
| Categorical distribution in value and reward modeling | True |
| The scale of supports used in categorical distribution | 300 |

Table 7: Key hyperparameters of **MuZero with/ SSL** on *Atari* environments.

| Hyperparameter | Value |
| --- | --- |
| Number of sampled actions (K) | Different environments have distinct values for K, shown in Figure 7 |
| Policy loss type | Cross Entropy Loss |

Table 8: Key hyperparameters of **Sampled EfficentZero** on *Atari* environments.

| Hyper-parameter | Value |
| --- | --- |
| Num of frames stacked | 1 |
| Num of frames skip | 1 |
| Reward clipping | False |
| Number of sampled actions (K) | 20 |
| Policy loss type | Cross Entropy Loss |
| Number of simulations in MCTS (sim) | 50 |
| Weight of policy entropy loss | 0.005 |
| Length of game segment | 200 |
| Use augmentation | False |
| Max gradient norm | 0.5 |

Table 9: Key hyperparameters of **Sampled EfficentZero** used in continuous control environments, such as, *MuJoCo*.

| Hyperparameter | Value |
|---|---|
| Num of max considered actions | i.e. Action Space Size |
| Gumbel Scale | 10 |
| Max visit init | 50 |
| Value Scale | 0.1 |

Table 10: Key hyperparameters of **Gumbel MuZero** on *Atari* environments.

| Hyperparameter | Value |
|---|---|
| Chance space size | i.e. Action Space Size |
| Afterstate Dynamics Network | Similar with Dynamics Network in Figure 24 |
| Afterstate Prediction Network | Similar with Prediction Network in Figure 24 |
| Chance Encoder | Two Layer MLP |

Table 11: Key hyperparameters of **Stochastic MuZero** on *Atari* environments.

| Hyperparameter | Value |
|---|---|
| Chance space size | $16 * num\_of\_possible\_chance\_tile$ |
| Afterstate Dynamics Network | Similar with Dynamics Network in Figure 24 |
| Afterstate Prediction Network | Similar with Prediction Network in Figure 24 |
| Chance Encoder | Two Layer MLP |
| Num of frames stacked | 1 |
| Num of frames skip | 1 |
| Reward clipping | False |
| Discount factor | 0.999 |
| Batch size | 512 |
| Length of game segment | 200 |
| TD steps | 10 |
| Use augmentation | False |
| Number of simulations in MCTS (sim) | 100 |

Table 12: Key hyperparameters of **Stochastic MuZero** on *2048* environments.

| Hyperparameter | Value |
|---|---|
| Board size | 6 |
| Num of frames stacked | 1 |
| Discount factor | 1 |
| Weight of policy loss | 1 |
| Weight of value loss | 1 |
| Number of simulations in MCTS (sim) | 100 |
| Categorical distribution in value modeling | False |

Table 13: Key hyperparameters of **AlphaZero** on *Gomoku* environments.

| Hyperparameter | Value |
|---|---|
| Board size | 6 |
| Num of frames stacked | 1 |
| Discount factor | 1 |
| Weight of SSL (self-supervised learning) loss | 0 |
| Length of game segment | 18 |
| TD steps | 18 |
| Use augmentation | False |
| Number of simulations in MCTS (sim) | 100 |
| The scale of supports used in categorical distribution | 10 |

Table 14: Key hyperparameters of **MuZero** on *Gomoku* environments.

