# OpenReview forum: "LightZero: A Unified Benchmark for Monte Carlo Tree Search in General Sequential Decision Scenarios"
_NeurIPS.cc/2023/Track/Datasets_and_Benchmarks — NeurIPS 2023 Datasets and Benchmarks Spotlight_

### Official Review · Reviewer_UFjT · 2023-07-20
**Review of LightZero**

**Rating:** 7
**Confidence:** 4
**Correctness:** Yes
**Clarity:** The paper is well written.

**Strengths:**

1. Clean implementations of almost all complicated MCTS/MuZero-like algorithms
2. Thorough analysis of the challenges of the general decision-making solvers
3. Extensive experiments and useful insights

**Additional Feedback:**

I appreciate the team's efforts and it is a good contribution to the community.

My suggestions for further improvements are:

1. Emphasizing Failure Cases for Algorithm Selection:
The paper presents an interesting analysis of RL algorithms for diverse environments, but it would significantly benefit from including a discussion about the failure cases. By understanding what makes certain algorithms unsuitable for specific environments, readers can gain valuable insights into algorithm selection and avoid potential pitfalls. Including discussions on both successful and unsuccessful cases would enable users to make more informed decisions, not only in selecting algorithms but also in tuning hyper-parameters and other settings.

2. Extending Analysis on the Impact of the Number of Players:
The paper mentions the inclusion of single-player, cooperative multi-player, and competitive multi-player environments, which is commendable. However, to improve the user's understanding of algorithm performance across different settings, the authors should delve deeper into the influence of the number of players on RL performance. For instance, how do algorithms behave in 2-player versus 4-player cooperative environments? Addressing such questions will provide readers with a more nuanced understanding of algorithmic behavior in various contexts.

3. In-Depth Exploration of Game Property Influence:
While the paper briefly mentions the cooperative and competitive properties of the environments, it would be beneficial to include a more comprehensive discussion on how these properties affect the performance of RL algorithms. The authors should elaborate on how cooperation and competition impact algorithm behavior, learning dynamics, and convergence rates. Additionally, analyzing how different RL algorithms handle specific game properties can provide users with practical insights to make informed choices for their respective applications.

**Documentation:**

The documentation is good.

**Limitations:**

The limitations are briefly discussed in the end of the main paper. But I would suggest the authors to add a more detail discussion of the limitation in the appendix.

**Opportunities For Improvement:**

I would suggest the authors to include more discussions about the failure cases. (I would discuss more in the additional feedback)

**Relation To Prior Work:**

Yes

**Summary And Contributions:**

This paper presents a clean and comprehensive implementation of almost all MCTS/MuZero-like algorithms. The authors provide a thorough analysis of the challenges of general MCTS-like decision-making solvers and a general treatment of the implementation of each modules in the algorithms. Extensive experiments demonstrate the correctness of the implementation and key insights are included in the paper for reader to follow when using this lib.

---

> ### Author Response · Authors · 2023-08-23
> **Rebuttal by Authors**
>
> Thank you for the valuable feedback and question. We will address your proposed improvements and limitations in the following parts:
>
> **Fail cases, limitations and future work**:  We sincerely agree to delve into the fail cases we encountered during our research process. Details such as network architecture design and hyper-parameter choice significantly affected the final performance. Below are some important practical details:
>   - Sparse reward environments: In these environments like minigrid [1], algorithms like EfficientZero that use the `value_prefix` prediction performed poorly, while MuZero with self supervised learning performed relatively better. We speculate that this might be due to most of the `value_prefix` values are zero in sparse reward. Using LSTM to predict these almost all-zero values might cause optimization problems, leading to performance degradation.
>   -  Stochastic dynamics environments: In the 2048 environment, due to the environment's intrinsic randomness, the performance of Stochastic MuZero was notably superior to MuZero. However, when we simply modified the 2048 environment by increasing the possible values of the tiles that appear, from {2,4} to {2,4,8,16,32}, the performance of both algorithms significantly declined. This phenomenon indicates that this randomness remains a significant challenge for MCTS-style algorithms. The detailed experiment results are shown in Figure 11 (Appendix).
>   -  Special network architecture: In Sampled EfficientZero algorithm applied to the lunarlander environment, we found that the residual connections and layer normalization in the network were crucial. If these modules were removed, the performance would significantly decrease.
>
> Besides, we also thoroughly discuss limitations and directions for future research. Hence, we have added a section “Limitations and Future Work” in the latest revised version of the supplementary materials.
>
> **The impact of the number of players**:
> As you pointed out, compared to single-agent environments, multi-agent environments face more significant challenges, including non-stationarity, joint state-action spaces, and credit assignment issues [2].
> In response to these challenges, our preliminary experiments mainly adopted the independent training. In this mode, we considered other agents as part of the environment, each agent making decisions independently while all agents shared the same network. During MCTS process, each agent also searches independently. We conducted experiments in the GoBigger environment [3] using custom T2P2, T2P3 and T2P4 modes (all other parameters remained consistent except for the number of players, P). The experimental results demonstrated that, no matter which mode, LightZero could achieve stable convergence in confrontations with bots, and its sample efficiency was 6 times higher than other methods, which confirmed that the performance did not suffer significant losses.
> These initial attempts have revealed the tremendous potential of data-efficient MCTS-style algorithms in multi-agent environments. We plan to provide baseline results of LightZero in more multi-agent environments (e.g. PettingZoo [4]). (Some work is in progress and can be found in our GitHub repository) Subsequently, we will delve into how to combine the characteristics of multi-agent environments for efficient MCTS search, and how to organically integrate some ideas from the MARL area [2][5], to design superior cooperative-competitive strategies, followed by related in-depth analysis.
>
> **In-depth exploration of game property influences**: In multi-agent problems, we can regard different game properties as distinct intrinsic randomness of environmental dynamics. This randomness will become a interesting challenge to tree-search-based planning methods for both efficiency and stability. Though our current experiments on a series of academic multi-agent envionments show LightZero can handle them well, we can imagine the potential and optimization space of MCTS-style methods to model massive agents and diverse agent's characters. And we will continue to work on this topic.
>
> [1] Chevalier-Boisvert, Maxime, et al. "Minigrid & Miniworld: Modular & Customizable Reinforcement Learning Environments for Goal-Oriented Tasks." arXiv preprint arXiv:2306.13831.
>
>
> [2] Rashid, Tabish, et al. "Monotonic value function factorisation for deep multi-agent reinforcement learning." The Journal of Machine Learning Research 21.1 (2020): 7234-7284.
>
> [3] Zhang, Ming, et al. "GoBigger: A Scalable Platform for Cooperative-Competitive Multi-Agent Interactive Simulation." The Eleventh International Conference on Learning Representations. 2022.
>
> [4] Terry, J., et al. "Pettingzoo: Gym for multi-agent reinforcement learning." Advances in Neural Information Processing Systems 34 (2021): 15032-15043.
>
> [5] Kuba, Jakub Grudzien, et al. "Trust region policy optimisation in multi-agent reinforcement learning." arXiv preprint arXiv:2109.11251 (2021).

---

### Official Review · Reviewer_AvYb · 2023-07-21
**Review for LightZero**

**Rating:** 6
**Confidence:** 3
**Correctness:** Yes.
**Clarity:** Yes.

**Strengths:**

I believe this is the first open-source library for learning-based MCTS methods. The authors certainly have put into much efforts in designing and implementing the library, taking into consideration of efficiency and effectiveness. The library has covered complete MCTS-RL algorithms and with a modular design, should support modular algorithm improvement.

A good amount of experiments have been conducted to verify the effectiveness of different design factors in the algorithms. While some findings are considered known, it is good to know the others.

**Additional Feedback:**

See Opportunities For Improvement.

**Documentation:**

Yes.

**Ethics:**

No.

**Limitations:**

No. No limitations are discussed. No Section 7 does not mention limitations.

**Opportunities For Improvement:**

I feel pretty conflicted in this work.

On one hand, the paper is, to the best of my knowledge, the first open-source complete library that implements the MCTS-style RL methods. This should be considered as a critical contribution to the research community, as they are known to be important, demonstrating effectiveness in high-profile journals, and yet actual working implementations are not open to the public.

However, on the other hand, the paper does not really make useful scientific discovery. Either some of the findings are already known to the community, or are not really guaranteed to be generalizable. Besides, LightZero seems to focus on the MuZero family, which has been more openly studied and accessible in the community. The more attractive side, AlphaZero, and related experiments to verify correctness are unfortunately missing.

So to sum up, I quite appreciate the efforts to make this library. However, the library does not advance our understanding of the algorithms, and neither does it hint at directions for further explorations, or problems to take a closer look at. I temporarily give a borderline reject rating and would sync with other reviewers for the final recommendation.

PS: Quantitative ratings in Figure 2 are pretty arbitrary. Please consider other ways to quantify on the dimensions, or make it qualitative only.

**Relation To Prior Work:**

Yes.

**Summary And Contributions:**

In this work, a new PyTorch-based computational framework for Monte Carlo Tree Search, called LightZero, is introduced. The framework / library covers the widely used learning-based tree search methods, including the AlphaZero family and the MuZero family. The library is designed with a modular approach such that each component could be separately studied. Using this library, the authors how various design factors impact the model performance. The experiments were conducted on MiniGrid to study exploration mechanism and selected Atari games to study alignment in environment model learning.

---

> ### Author Response · Authors · 2023-08-22
> **Rebuttal by Authors**
>
> Thank you for the valuable feedback and question. We will address your problems in the following parts:
>
> **Useful scientific discovery**: We greatly appreciate your endorsement that LightZero is the first open-source complete library implementing MCTS-style RL methods. Building upon this framework, we also proposed some techniques in original paper to make LightZero more powerful and faster than previously proposed algorithms. For example, in section 5.2 and Appendix D.2, we analyze the alignment problem in environment model learning. Specifically, MuZero [1] does not provide any explicit supervision about the learning process of  the representation network for generating hidden state, while some other model-based RL methods [2][3] utilized  the Recurrent State-Space Model (RSSM) and predictors to learn a more accurate world model. EfficientZero [4] and ROSMO [5] tried to relieve this problem for MuZero, but they only conducted experiments on Atari environments. LightZero extends the idea of the consistency loss used in EfficientZero and provides an in-depth analysis of its effectiveness on three different types of environments (Atari, LunarLander and board games) — the inappropriate use of the consistency loss will lead to little performance gain (Figure 6) or even drops (Figure 6 in Appendix). Another novel insight for the efficiency optimization is described in Appendix F (especially Figure 7 and Figure 8), we have proposed two important aspects for accelerating MCTS-style RL training: parallel (batched) data collecting pipelines and peer-to-peer communication with the RDMA technique. Besides, the special exploration mechanism in MCTS (mentioned in Section 5.1 and Appendix D.1) and key observations (mentioned in Appendix C) can also be considered as some useful thoughts and information for other practical applications.
>
> **AlphaZero family**: For better application of MCTS+RL algorithms in different domains, we agree that both MuZero and the AlphaZero family are important and necessary. However, we want to clarify that we have verified the correctness of AlphaZero-type algorithms in the original paper (Gomoku experiment in Figure 4). To further support and expand the AlphaZero family in LightZero, we first deployed a tuned AlphaZero in a series of board games, including TicTacToe, Connect4 and 9x9 Go. (Detailed experiments and code implementations can be found in our GitHub repository). Moreover, we also explore and extend two variants: AlphaZero with the sampled action mechanism (similar to Sampled MuZero) and AlphaZero with the gumbel tree search mechanism. The former is used to solve the complex action space problem in AlphaDev [6] and the latter can also reduce the number of simulations in AlphaZero (mentioned in [7]). We plan to incorporate all of these improvements and add detailed hyper-parameter settings in the revised version of the paper.
>
> **Quantitative ratings in Figure 2**: We thank you for pointing out the presentation issue of Figure 2, which might result in some kinds of misunderstandings. However, we need to emphasize that Figure 2 is already qualitative rather than quantitative ratings. Concretely, we have already added the detailed explanation for Figure 2 in Appendix E of supplementary material, showing the qualitative criteria for different levels (0-5). We will further refine the caption and the graph of Figure 2 to make it clearer.
>
> Thank you for the valuable feedback, we are committed to enhancing the quality and completeness of our study by addressing the above problems in the revised version of the paper.
>
> [1] Schrittwieser, Julian, et al. "Mastering atari, go, chess and shogi by planning with a learned model." Nature 588.7839 (2020): 604-609
>
> [2] Hafner, Danijar, et al. "Mastering atari with discrete world models." arXiv preprint arXiv:2010.02193 (2020).
>
> [3] Hafner, Danijar, et al. "Mastering diverse domains through world models." arXiv preprint arXiv:2301.04104 (2023).
>
> [4] Ye, Weirui, et al. "Mastering atari games with limited data." Advances in Neural Information Processing Systems 34 (2021): 25476-25488.
>
> [5] Liu, Zichen, et al. "Efficient Offline Policy Optimization with a Learned Model." arXiv preprint arXiv:2210.05980 (2022).
>
> [6] Mankowitz, Daniel J., et al. "Faster sorting algorithms discovered using deep reinforcement learning." Nature 618.7964 (2023): 257-263.
>
> [7] Danihelka, Ivo, et al. "Policy improvement by planning with Gumbel." International Conference on Learning Representations. 2021.

---

> > ### Comment · Reviewer_AvYb · 2023-08-23
> >
> > I have read the Appendix on Figure 2 but still the ratings are pretty arbitrary and I believe it is still hard to quantitatively score the algorithms. Please reconsider the way for presentation.
> >
> > In terms of the AlphaZero family, I'm concerned if the experiments on Gomoku suffice.
> >
> > More importantly, I recently knew of MCTX, the JAX library for MCTS-based RL, which precedes this work, so my comments on the work being the first public implementation are actually wrong.
> >
> > All being said, I tend to believe the PyTorch implementation still has merits as it's more widely used than JAX. I would like to raise my score to 6.

---

> > > ### Author Response · Authors · 2023-08-24
> > > **Response about the comparison between LightZero and MCTX**
> > >
> > > LightZero is an algorithm benchmark implemented based on PyTorch, integrating nine types of algorithms such as AlphaZero and MuZero, and supporting over twenty environments, including Atari and Go. On the other hand, the MCTX library, primarily implemented based on JAX, incorporates basic implementations of algorithms such as AlphaZero, MuZero, and Gumbel MuZero, but has yet to finalize the training process across various environments.
> > >
> > > To intuitively compare the differences in integrated algorithms and supported environments between these two repositories, we have displayed the algorithms and environments supported by LightZero and MCTX in two tables below.
> > >
> > > **Note**:
> > > 1. "✔" indicates that the corresponding item has been completed and thoroughly tested.
> > > 2. "🔒" signifies that the corresponding item is currently in progress.
> > > 3. "---" indicates that the algorithm does not support this environment.
> > >
> > > `LightZero`
> > >
> > > | Algo \\ Env | ClassicControl | Box2D | Atari | MuJoCo | GoBigger | MiniGrid | Maze | ConnectFour | Gomoku | 2048 | Go  |
> > > | --------------------------------------------------------- | --------------- | ----- | ----- | ------ | -------- | -------- | ---- | ----------- | ------ | ---- | --- |
> > > | AlphaZero                                                 | ---             | ---   | ---   | ---    | ---      | 🔒        | 🔒   | ✔           | ✔      | 🔒   | ✔   |
> > > | Sampled AlphaZero                                         | ---             | ---   | ---   | ---    | ---      | 🔒        | 🔒   | 🔒          | ✔      | 🔒   | ✔    |
> > > | Gumbel AlphaZero                                          | ---             | ---   | ---   | ---    | ---      | 🔒        | 🔒   | 🔒          | ✔      | 🔒   | ✔   |
> > > | MuZero                                                    | ✔               | ✔     | ✔     | ---    | ✔        | ✔         | ✔   | ✔          | ✔      | ✔   | ✔   |
> > > | MuZero w/ SSL                                             | ✔               | ✔     | ✔     | ---      | ✔        | ✔         | 🔒   | 🔒          | ✔      | ✔   | ✔   |
> > > | EfficientZero                                             | ✔               | ✔     | ✔     | ---       | ✔        | ✔         | 🔒   | 🔒          | ✔      | ✔   | ✔   |
> > > | Gumbel MuZero                                             | ✔               | ✔     | ✔     | ---      | ✔        | ✔         | 🔒   | 🔒          | ✔      | ✔   | ✔   |
> > > | Sampled MuZero                                            | ✔               | ✔     | ✔     | ✔      | ✔        | ✔         | 🔒   | 🔒          | ✔      | ✔   | ✔   |
> > > | Stochastic MuZero                                         | ✔               | ✔     | ✔     | ---      | ✔        | ✔         | 🔒   | 🔒          | ✔      | ✔   | ✔   |
> > >
> > > `MCTX`
> > >
> > > | Algo \\ Env | ClassicControl | Box2D | Atari | Maze | ConnectFour | Gomoku | Go  |
> > > | ---------------------------------------------------- | -------------- | ----- | ----- | ---- | ----------- | ------ | --- |
> > > | AlphaZero                                            | ---            | ---   | ---   | ✔    | ✔           | ✔      | ✔   |
> > > | MuZero                                               | ✔              | ✔     | 🔒    | 🔒   | 🔒           | 🔒      | 🔒   |
> > > | Gumbel MuZero                                        | 🔒             | 🔒     | 🔒   | 🔒   | 🔒           | 🔒      | 🔒   |
> > > | Stochastic MuZero                                    | 🔒             | 🔒     | 🔒   | 🔒   | 🔒           | 🔒      | 🔒   |
> > >
> > > It's important to note two aspects: Firstly, the algorithms and environments supported by MCTX listed below not only encompass those of MCTX itself but also extend to all repositories derived from MCTX. Secondly, all environments associated with Gumbel MuZero and Stochastic MuZero are currently in a locked 🔒 state. To the best of our knowledge, neither MCTX nor its derivative repositories have entirely implemented these algorithms and associated environments. While the fundamental modules for the Gumbel MuZero and Stochastic MuZero algorithms have been established in the original MCTX repository, the development of a comprehensive training pipeline for these algorithms is still in progress.

---

### Official Review · Reviewer_dyMN · 2023-07-21
**Useful contribution to the decision-making community**

**Rating:** 7
**Confidence:** 3
**Correctness:** The claims seem correct

**Strengths:**

MCTS style approaches are clearly one of the most promising set of approaches for sequential planning problems, so creating a unified structure for comparing and implementing the latest approaches is a useful contribution to the decision-making community

The comparisons done across algorithms are quite interesting, especially results that were only included in the appendix.


**Additional Feedback:**

It is not clear to me that the present work can be considered a benchmark (as it is re-using common RL benchmark tasks), but is rather a framework for implementing MCTS style algorithms. It is true that you have used this framework to implement and compare a variety of MCTS algorithsm (which is very useful) but I don't know how will it matches the intention of this track. I will leave it to the Area Chair to pass judgement on this.

**Clarity:**

The paper has a number of clarity issues that make it difficult to understand on the first readthrough. As previously mentioned there are a lot of grammatical mistakes and typos.

Beyond that, however, the contribution of this work is sometimes unclear. At times (like in figure 2) it seems like LightZero is an algorithm, when in fact it is a framework for implementing MCTS-style agorithms. Also calling it a benchmark sort of seems misleading as well (see my note on additional feedback).

**Documentation:**

Yes this seems sufficient

**Ethics:**

No concerns

**Limitations:**

One key point that is worth clarifying in the next version of this paper is how well do you implementation of these 9 algorithms compare to the implementations in their respective papers? Were you able to make 1-1 comparisons to ensure that all features were implemented and tuned properly? Were you able to achieve the same or higher performance than that quoted in the original work.

**Opportunities For Improvement:**

I think the main opportunities for improvement are around the clarity of the presentation of this work (see section on clarity)

Also, I personally would like to see more time spent on summarizing these interesting results in the main body of the paper.

There are numerous grammatical and spelling mistakes and the manuscript should be given a thorough proofread.
A few examples are
On line 52 “to outcome” should be “to overcome”
On lines 57-58 I don’t understand the sentence “its internal procedure can be deeply optimized without any interference”
On line 115  “decouplement” is incorrect
On line 196 “Environement” is spelled wrong

**Relation To Prior Work:**

Yes this is made clear

**Summary And Contributions:**

This work introduces a unified framework for designing and deploying MCTS-based reinforcement learning algorithms called LightZero. It is designed in a modular way with four main components: Data collector, Data arranger, agent learner, and agent evaluator. 9 recent MCTS-style algorithms are re-implemented in this framework and compared across a variety of RL tasks. Various design choices are compared across tasks.

---

> ### Author Response · Authors · 2023-08-23
> **Rebuttal by Authors**
>
> Thank you for the valuable feedback. We will address your comments and concerns in the following parts:
>
> **Clarity**: we are grateful to your detailed comments about the grammatical and spelling mistakes in the original paper, and we are committed to enhancing the quality and completeness of our work. Also, we polish the mixed usage of LightZero as framework/benchmark/algorithm and add necessary explanation in corresponding paragraph. More experimental figures and description are also added into the appendix. All the above modifications has been marked with red color in the latest revised version of the paper. For suggestions about more discussions about interesting observations and insights in the main paper, we will rearrange the entire representation structure and in camera-ready phase if accepted.
>
> **Explanation about LightZero as a framework/benchmark/algorithm**: LightZero is the first general MCTS/MuZero algorithm benchmark. In this paper, we introduce a series of framework designs to decouple sophisticated pipelines of MCTS-style methods, then we reimplement and refine existing related research works, tuning these algorithm variants and summarizing several key observations and insights, which further systematically evaluate related algorithms and system design. Not only current methods can be benchmarked by LightZero, but also the newly proposed MCTS-style methods can be easily integrated and evaluated in LightZero. Specifically, to underscore the disparity between the individual MCTS+RL algorithms, each engineered to overcome particular challenges, and a universally robust MCTS algorithm, we've called an special algorithm within our current benchmark as LightZero. This algorithm represents a potential optimal blend of diverse techniques and hyperparameters. However, the discovery of this optimal algorithm on all the dimensions remains an ongoing quest. Our objective, by contrasting LightZero with other methods, is to reveal potential avenues for future research enhancements.
>
>
> **1-1 comparisons**: We agree with the necessity of comparisons between LightZero and respective algorithms to ensure that all features were implemented and tuned properly. However, most algorithms in the MCTS+RL area often conducted experiments on some special environments or tested in a large number of environmental interactions, which limits reproducibility and accessibility. Consequently, we plan to conduct extensive ablation analysis experiments in a range of academic reinforcement learning environments, encompassing those mentioned in previous literature as well as other well-established environments. These experiments aim to evaluate the impact of each algorithmic design, feature, and technique, and by deeply dissecting the operational mechanisms of MCTS+RL, we aspire to discover a more efficient, robust, and universally applicable decision-making algorithm.

---

> > ### Comment · Reviewer_dyMN · 2023-08-23
> > **Thanks for the comments**
> >
> > I have read through the comments and responses. I still think it is confusing to call the framework and the algorithm "LightZero". I agree with some other revewiers comments that more extensive experiments on the more challenging benchmarks would be a large benefit to this already good work. I will keep my score as is.

---

### Official Review · Reviewer_8SGB · 2023-07-21
**Overall good paper on benchmarking various RL-style approaches that use MCTS**

**Rating:** 7
**Confidence:** 5
**Correctness:** Yes, the paper's claims are correct.

**Strengths:**

MCTS is widely used in several RL-based approaches, but no unified study can quantify and evaluate these algorithms on standard benchmarks. This paper addresses these needs and decouples the key components of such algorithmic approaches. Decoupling the essential components also provides insights into how MCTS-based approaches can be potentially used for real-world problems, which has been lacking so far, with existing applications often limited to board games.

**Additional Feedback:**

N/A

**Clarity:**

The paper is well-written. However, the details of the framework and the sub-modules could be clarified significantly.

**Documentation:**

Do the authors envision releasing this framework for future benchmarking? If yes, the documentation on how to use the framework is lacking.

**Limitations:**

See above.

**Opportunities For Improvement:**

I would encourage the authors to first decouple the key components of MCTS without referencing RL-based approaches. For example, if this framework is meant to be used for benchmarking future studies as well, how do users encode generative models, model-based rollouts, etc., in the framework? The paper focuses a lot on "learning," which is extrinsic to MCTS per se, as MCTS is a fully online approach to planning.

Second, the four key sub-modules need to be described at a significantly greater depth. Note that this categorization of the modules is new, i.e., it is not a well-known representation of decoupling the operation of MCTS-based RL approaches. As a result, it is not immediately clear how the sub-modules map to different algorithms. The authors can choose a specific algorithm, e.g., AlphaZero, and describe its components mapped to the sub-module in detail.

While the application of MCTS is limited to real-world problems, there are many real-world examples, e.g., robot optimization in factories (Baier et al., AAMAS), and emergency response (Mukhopadhyay et al., AAMAS). I would also encourage the authors to benchmark the algorithms on some real-world problems. While this is not a deal-breaker, I think this would make the paper more self-complete.

Typos: "Exploration" is incorrectly spelled on page 7, section 5.1, first paragraph.

**Relation To Prior Work:**

Yes, to the best of my knowledge, this is the first unified framework for MCTS-based RL approaches.

**Summary And Contributions:**

The paper presents a framework for benchmarking different RL-style approaches that use Monte Carlo tree search for sequential decision-making under uncertainty. While the paper includes several algorithms and environments, I have questions about the reusability of the framework for future benchmarking.

---

> ### Author Response · Authors · 2023-08-22
> **Rebuttal by Authors (Part I)**
>
> Thank you for the valuable feedback and suggestions. We will address your issues in the following parts:
>
> **Combination with future studies**: We strongly agree with the importance of ease of integration with LightZero and other future research. Here we describe two examples about future directions and qualitative implementation costs.
> - The first one is to utilize the Large Language Model (LLM). On the one hand, LLM can be viewed as a high-level task planner, which breaks down a complex decision-making problem into a series of low-level instructions [1], LightZero can set these instruction as conditional inputs of the representation network and serve as the powerful instruction executor. On the other hand, to leverage the ideal plan proposed by LLM and the actual behaviour in practice, it is necessary to guide the decoding strategies of LLM to adapt the interaction results of low-level agents and the environment. Ground Decoding [2] propose to utilize RL agents as "grounded models" to guide the probability of LLM's output tokens. MCTS-style RL agents can provide long-horizion and more stable grounded signals due to its tree search planning properties. The concrete implementation is quite simple because it only needs to combine the searched policy action selection probability and LLM's output. Besides, RL with Human Feedback (RLHF) [3] has already emerged as an important technique to finetune LLM and related alignment tasks. Given that RLHF often needs continual online iterations, current RL methods are mainly of the on-policy types (e.g. PPO, A2C) . However, MuZero Unplugged [4] with the powerful reanalyze mechanism can seaminglessly deal with online and offline RL cases, which exhibits potential on the problem of incremental training over multiple iterations. Therefore, we also consider that it may also be a possible choice for RLHF.
> - The second one involves greater use of model-based RL techniques. LightZero has already integrated many important model-based RL methods, such as Recurrent State-Space Model (RSSM) used in Dreamerv2 [5], hidden state consistency loss used in EfficientZero [6], model ensemble tricks used in STEVE [7], complicated afterstate and related dynamics functions proposed in Stochastic MuZero [8] and so on. Other methods can also be implemented in the dynamics network and the loss function without significant disruption to the overall pipeline.
>
> **Detailed sub-modules description of MuZero**: To better illustrate how a specific algorithm is mapped into different sub-modules in LightZero, we describe the implementation of MuZero algorithm from viewpoints of sub-modules mentioned in Figure 3:
>
> - Data Collector: MuZero is a kind of online RL algorithm, so it requires numerous interactions between the agent and the environment to collect training data. We utilize the vectorized environment manager and the deeply optimized batch-parallel search tree via Cython/C++ extensions to ensure high-throughput data collection.
> - Data Arranger: In order to make full use of different stale data, in this module MuZero needs to first store the entire trajectories or episodes in the prioritized replay buffer so as to keep their temporal sequence. Then we can employ the priority recomputer mechanism to periodically calculate the sampling priority for these data, which is proportional to the probability that data will be sampled for training. Moreover, to correct the off-policy bias and make the training more stable, the data reanalyzer is used to update the target value stored in data with the latest network parameters. To balance the complicated "producer-consumer" relatinship between data collector and agent learner, we also design a throughput limitier in MuZero to monitor the number of push/pop data operations and further control corresponding computation resource allocation.
> - Agent Learner: This module contains various deep learning and reinforcement learning techiniques to train a group of neural networks defined in MuZero, such as the distributional RL module to model the intrinsic randomness of environmental reward, the data parallel and mixed precision training utilities to accelerate the per iteration time. All of these options can be easily enabled/disabled by the corresponding configuration fields.
> - Agent Evaluator: Throughout the training process, MuZero needs to evaluate the performance of the newly trained network. This module not only involves some evaluation tricks like low temperature sampling and beam search to improve results, but also uses a lot of metrics and visualizations to analyze agent's behaviours.
> - Context Exchanger: To achieve the asynchronous execution of the above four modules and the ability to efficiently scale the entire training pipeline, a context exchange with some novel communication components is used to efficiently transfer the necessary context information.
>
> The other rebuttal parts and the reference part can be found in next comment.

---

> ### Author Response · Authors · 2023-08-22
> **Rebuttal by Authors (Part II)**
>
> Here we continue to discuss the above rebuttal parts:
>
> **Real-world application**: For the problem of the real-world application, The GoBigger environment mentioned in the main paper (Figure 4) can be regarded a good application example. GoBigger is inspired by the popular online multiplayer game Agar.io [9], which has more than 100 million downloads. On this basis, many modifications have been made to allow more cooperation and competition in GoBigger. We evaluated GoBigger 2 Team 2 Player (T2P2) mode in the main paper and also verfity its performance on modes with more players (T2P3 and T2P4). More results can be found in GitHub repository. Besides, we also try to apply LightZero on the management problem of unmanned aerial vehicle (UAV) swarm. Our another paper "Scheduling UAV Swarm with Attention-based Graph Reinforcement Learning for Ground-to-air Heterogeneous Data Communication" (camera-ready phase now) combines an attention-based graph and LightZero framework to efficiently allocate resources to mobile sensors in data-heterogeneous environments, taking into account the constraints of drone communication capabilities and battery life constraints. For future research, we will consider more applications in the field of computer system optimization, similar to AlphaDev [10].
>
> **Release and documentation**: We have already released the main part of LightZero and you can access the whole framework at this GitHub [url](https://github.com/opendilab/LightZero). We report most benchmark results in [README.md](https://github.com/opendilab/LightZero#readme). We are also working on improving the documentation, such as ["How to transfer a new decision-making environment into LightZero"](https://github.com/opendilab/LightZero/blob/main/docs/source/tutorials/envs/customize_envs.md) and ["How to add new algorithm techniques in LightZero"](https://github.com/opendilab/LightZero/blob/main/docs/source/tutorials/algos/customize_algos.md).
>
> We are grateful for your valuable input and thorough evaluation of our work. and we are committed to enhancing the quality and completeness of our study by addressing above problems and presentation drawbacks (including typos and missing details) in the revised version of the paper.
>
> [1] Huang, Wenlong, et al. "Language models as zero-shot planners: Extracting actionable knowledge for embodied agents." International Conference on Machine Learning. PMLR, 2022.
>
> [2] Huang, Wenlong, et al. "Grounded decoding: Guiding text generation with grounded models for robot control." arXiv preprint arXiv:2303.00855 (2023).
>
> [3] Touvron, Hugo, et al. "Llama 2: Open foundation and fine-tuned chat models." arXiv preprint arXiv:2307.09288 (2023).
>
> [4] Schrittwieser, Julian, et al. "Online and offline reinforcement learning by planning with a learned model." Advances in Neural Information Processing Systems 34 (2021): 27580-27591.
>
> [5] Hafner, Danijar, et al. "Mastering atari with discrete world models." arXiv preprint arXiv:2010.02193 (2020).
>
> [6] Ye, Weirui, et al. "Mastering atari games with limited data." Advances in Neural Information Processing Systems 34 (2021): 25476-25488.
>
> [7] Antonoglou, Ioannis, et al. "Planning in stochastic environments with a learned model." International Conference on Learning Representations. 2021.
>
> [8] Buckman, Jacob, et al. "Sample-efficient reinforcement learning with stochastic ensemble value expansion." Advances in neural information processing systems 31 (2018).
>
> [9] https://play.google.com/store/apps/details?id=com.miniclip.agar.io&hl=en_US&pli=1
>
> [10] Mankowitz, Daniel J., et al. "Faster sorting algorithms discovered using deep reinforcement learning." Nature 618.7964 (2023): 257-263.

---

> > ### Comment · Reviewer_8SGB · 2023-08-23
> > **Reviewer Feedback**
> >
> > Thank you for the detailed response. It answered my queries. I have updated the original review based on this response.

---

### Official Review · Reviewer_xvyT · 2023-07-21
**very good paper, supplementary material answered 90% of my questions**

**Rating:** 9
**Confidence:** 5
**Correctness:** Yes
**Clarity:** Yes

**Strengths:**

* This library  will significantly impact on implementation of the future MuZero algorithms for the research community.
* The algorithm and system design of MCTS methods and design a modular training pipeline which can easily integrate novel insights for better scalability.
* Algorithm performance was one of the main considerations. For example they made very deepdive into the MCTS implementation.
* All most popular MCTS Model Based algorithms were supported.


**Additional Feedback:**

I'd like to see more long benchmarks on the complex envs.
Also there was a good explanation about MCTS implementation issues but I'd liek to see a full end2end gpu only pipeline in the future.

**Documentation:**

Github repository has pretty good documentation and it was easy for me to run some of their experiments.

**Ethics:**

I don't see any ethical concerns.

**Limitations:**

I don't see any negative societal impact.

**Opportunities For Improvement:**

* Run much longer tests.  For example atari pacman have been trained on 0.5m steps didn't even achieve noticeable scores. 2500 is too low. Same issue with breakout too.  It is not challenging to achieve near 400 scores  but go thru the first level to the 800+ is much more interesting but experiment was stopped.
* Run experiments on the more complex environments. For example BipedalWalker-v3 in 5m steps did not achieve 300+ scores looks surprisedly bad compared to the atari results but this env could be easily solved in less steps using even PPO. Hardcore version looks much more interesting.
* Run big multigpu/multinode training on selfplay or really complex environment. For example 9x9 go should be achievable.


**Relation To Prior Work:**

Yes

**Summary And Contributions:**

* Clean and easy to use pytorch implementation of the *Zero algorithms family.
* Provided deep analysis of almost all existing algorithms.
* Analysis covered a broad range of the different environments: both vision and state observations and discrete and continuous action spaces.
* list of the supported algorithms is really impressive:
- AlphaZero
- MuZero
- MuZero reanalyze
- Efficient MuZero
- Sampled MuZero
- VQVAE MuZero
- Gumbel MuZero
- Stochastic MuZero

---

> ### Author Response · Authors · 2023-08-24
> **Rebuttal by Authors**
>
> Thank you for your valuable feedback. We will address your comments and concerns in the following sections.
>
> **Longer runs and MsPacman/Breakout results**:
>   - Following your suggestions, we conducted baseline experiments over 2M environment steps in the Atari `MsPacman` and `Breakout` settings. The table below illustrates the average returns plus standard deviations of different algorithms across five test episodes.
>
>     | Algo. / Env | MsPacman | Breakout |
>     |-------------|---------|----------|
>     | MuZero | 2100 ± 40 | 38 ± 56 |
>     | MuZero w/ SSL | 2986 ± 35 | 532 ± 98 |
>     | EfficientZero | 2453 ± 45 | 486 ± 56 |
>     | Sampled EfficientZero | 2019 ± 86 | 398 ± 65 |
>
>   - In the `MsPacman` environment, we observed a near-monotonic increase in average test rewards as the training steps increased, indicating that our model maintains effective learning over extended training periods. On the `Breakout `setting, while the model performance appeared to saturate after reaching a certain peak, we believe this may be due to substantial variations across different levels in the Breakout environment. To further validate the stability of our implemented algorithm, we will add long-term training results on additional environments in the revised manuscript.
>   -  It's worth noting that the MuZero series of algorithms are usually compared on fewer data interactions (e.g., 100K env steps). Therefore, our main experiments are conducted at lower training steps. Compared with the results in the Table 1 of the original EfficientZero paper [1], our EfficientZero implementation, at the same training steps, reached close to 3000 points in MsPacman (1400 points in [1]), and close to 550 points in Breakout (400 points in [1]). These results show that our implementation has reached or exceeded the performance level of the original paper.
>
> **More complex environments**:
>   - On the issue of real-world applications: The GoBigger environment mentioned in the main paper (Figure 4) can be considered as a good application example. GoBigger is inspired by the popular online multiplayer game Agar.io, which has more than 100 million downloads. We evaluated the GoBigger 2 Team 2 Player (T2P2) scenario in the main paper and also verified its performance in scenarios with more players (T2P3). More results can be found in the GitHub repository. In addition, we also attempted to apply LightZero to the management problem of unmanned aerial vehicle (UAV) swarms. Our other paper, "Scheduling UAV Swarm with Attention-based Graph Reinforcement Learning for Ground-to-air Heterogeneous Data Communication" (currently in the camera-ready phase), combines an attention-based graph and LightZero framework to efficiently allocate resources to mobile sensors in data-heterogeneous environments. In future research, we will consider more applications in the field of computer system optimization, similar to AlphaDev [4].
>   - For BipedalWalker-v3 experiment: Based on your suggestion, we tested the performance of Sampled EfficientZero on the BipedalWalker-v3 Hardcore version and achieved around $280 \pm 48$ points at about 5M Env Steps. It's worth noting that since MCTS search is mainly performed on discrete actions, the high-dimensional continuous action space has always been a major challenge for MCTS series algorithms. For example, as pointed out in Figure 4 of the Sampled MuZero paper [2], in the 4-dimensional continuous action space environment DMC environment `hopper.hop`, it also takes more than 5M Env Steps for Sample MuZero and D4PG [3] and other algorithms to converge. This implies that our implementation has achieved similar performance as the recent Sampled MuZero paper.
>
> **Multi-GPU/Multi-node training**:
>   - By using the Distributed Data Parallel module of PyTorch, we have added support for multi-GPU training in our code repository and have tested the EfficinetZero on the Atari Pong environment with 1/2/4 GPUs. The test speed is shown in the table below, and the results show that training with 4 GPUs can improve the speed by about 5 times with similar performance, which is in line with our expectations. We are currently conducting self-play training within a 9x9 Go environment, details of which can be found in our repository.
>
>
>     Number of GPUs | Approximate Time to 1M Env Steps
>     ----- | -----
>     1    | 844 mins
>     2    | 363 mins
>     4    | 152 mins
>
>
> [1] Ye, Weirui, et al. "Mastering atari games with limited data." Advances in Neural Information Processing Systems 34 (2021): 25476-25488.
>
> [2] Hubert, Thomas, et al. "Learning and planning in complex action spaces." International Conference on Machine Learning. PMLR, 2021.
>
> [3] Barth-Maron, Gabriel, et al. "Distributed distributional deterministic policy gradients." arXiv preprint arXiv:1804.08617 (2018).
>
> [4] Mankowitz, Daniel J., et al. "Faster sorting algorithms discovered using deep reinforcement learning." Nature 618.7964 (2023): 257-263.

---

### Author Response · Authors · 2023-08-23
**General Response**

We appreciate the valuable insights and suggestions provided by the reviewers. Guided by your feedback, we have executed the following general enhancements to our manuscript, which have been highlighted in red in the revised version of the paper:
- We have rectified the grammatical errors throughout the text and clarified areas where the expression was unclear.
- We have revised the references pertaining to "Standardization and Reproducibility" in the main text to align more closely with the theme of MCTS + RL.
- In Appendix Section I.2.4 Stochastic MuZero Benchmark, we added more baseline experiments for Stochastic MuZero and MuZero in 2048 environments with varying levels of stochasticity. These experiments validate the advantage of Stochastic MuZero over MuZero in environments with inherent randomness, while also highlighting the challenges posed to MCTS algorithms as stochasticity increases.
- We included a new section in the appendix, I.2.6 Multi Agent Benchmark, where we discuss our experimental setup in the multi-agent environment gobigger and analyze future research directions.
- We have also added a new section on "Limitations and Future Work" in the appendix, where we discuss the limitations of LightZero when applied to complex real-world problems and point out areas for future exploration.
- Documentation:  We are working on refining the comprehensive user documentation for LightZero, which includes guidelines on how to customize environments and algorithms within LightZero. We will continue to enhance the completeness and practicality of these documents in subsequent updates.

We hope these revisions address with your concerns, and we extend our gratitude for your precious suggestions. We eagerly await your further discussion and feedback.

---

### Decision · Program_Chairs · 2023-09-22

**Decision:**

Accept (Spotlight)

**Comment:**

The paper describes a new open source library to design Monte Carlo Tree Search (MCTS)algorithms.  The library is modular and it includes many existing and sophisticated MCTS algorithms.  This is important contribution to the community that will help further advances for sequential decision making based on online planning.